# Molecular architecture of chitin and chitosan-dominated cell walls in zygomycetous fungal pathogens by solid-state NMR

Qinghui Cheng [1,5], Malitha C. Dickwella Widanage[1,4,5], Jayasubba Reddy Yarava [1], Ankur Ankur[1], Jean-Paul Latgé[2], Ping Wang [3] & Tuo Wang [1] ✉

Zygomycetous fungal infections pose an emerging medical threat among individuals with compromised immunity and metabolic abnormalities. Our pathophysiological understanding of these infections, particularly the role of fungal cell walls in growth and immune response, remains limited. Here we conducted multidimensional solid-state NMR analysis to examine cell walls in five Mucorales species, including key mucormycosis causative agents like *Rhizopus* and *Mucor* species. We show that the rigid core of the cell wall primarily comprises highly polymorphic chitin and chitosan, with minimal quantities of β-glucans linked to a specific chitin subtype. Chitosan emerges as a pivotal molecule preserving hydration and dynamics. Some proteins are entrapped within this semi-crystalline chitin/chitosan layer, stabilized by the sidechains of hydrophobic amino acid residues, and situated distantly from β-glucans. The mobile domain contains galactan- and mannan-based polysaccharides, along with polymeric α-fucoses. Treatment with the chitin synthase inhibitor nikkomycin removes the β-glucan-chitin/chitosan complex, leaving the other chitin and chitosan allomorphs untouched while simultaneously thickening and rigidifying the cell wall. These findings shed light on the organization of Mucorales cell walls and emphasize the necessity for a deeper understanding of the diverse families of chitin synthases and deacetylases as potential targets for novel antifungal therapies.

Mucormycosis, a severe infection caused by molds belonging to the order of Mucorales, predominantly manifests in individuals with a diverse range of metabolic abnormalities, including inadequately controlled diabetes mellitus, acidosis, acquired iron overload syndromes, immunometabolism dysregulation triggered by critical illness, and immunosuppression stemming from transplantation[1,2]. Amid the coronavirus pandemic, mucormycosis has emerged as a prevalent co-infection, identified as COVID-19-associated mucormycosis (CAM)[3,4]. This is attributed to immune dysregulation caused by COVID-19 and treatments such as steroids, which compromise the

[1]Department of Chemistry, Michigan State University, East Lansing, MI, USA. [2]Institute of Molecular Biology and Biotechnology, University of Crete, Heraklion, Greece. [3]Department of Microbiology, Immunology and Parasitology, Louisiana State University Health Sciences Center, New Orleans, LA, USA. [4]Present address: Renewable Resources and Enabling Sciences Center, National Renewable Energy Laboratory, Golden, CO, USA. [5]These authors contributed equally: Qinghui Cheng, Malitha C. Dickwella Widanage. ✉e-mail: wangtuo1@msu.edu

body's ability to resist fungal infections[3]. The outbreak of pulmonary and rhino-orbital cerebral mucormycosis in India during the early 2021 second wave of COVID-19 was particularly notable[5,6]. Mucormycosis is predominantly caused by *Rhizopus* species, responsible for over half of all diagnosed cases, followed by *Mucor* species[7,8]. The current first-line treatment involves liposomal amphotericin B (AmB), supplemented with second-line agents like isavuconazole or posaconazole, and it often requires aggressive surgical removal of infected tissue prior to antifungal treatment[9].

Although mucormycosis exhibits relatively low morbidity, its mortality rate remains over 40% even after treatment and nearly 100% in patients with disseminated disease, neutropenia, or brain infections[10,11]. Due to the low efficacy of existing azole and polyene therapies, ongoing efforts should focus on developing novel agents, especially those designed to target the fungal cell wall, which is a vital organelle regulating structural integrity[12,13]. Mucorales are insensitive to the only class of antifungal drugs targeting the cell wall—specifically, the echinocandins that inhibit β−1,3-glucan synthesis[14–16]. This insensitivity may stem from key structural differences between the cell walls of Zygomycetes (including Mucorales) and those found in Ascomycetes and Basidiomycetes[17].

Early chemical analyses of Mucorales unraveled a unique composition of their cell walls, identifying chitin and chitosan as the primary carbohydrate polymers[17]. N-acetylglucosamine (GlcNAc) and glucosamine (GlcN), the constituent sugar units forming chitin and chitosan, were found to represent 34–57% of the cell walls in *R. oryzae* and *R. delemar*[18,19]. Intriguingly, β-glucans, the predominant polysaccharides in most fungal species, were found to constitute only 3–12% of the cell wall polysaccharides in these *Rhizopus* species. This discrepancy may contribute to the inefficiency of echinocandins against Mucorales. It also prompts questions about how the structural integrity of the cell wall is maintained in the absence of this crucial molecule, which plays a pivotal role in cell wall crosslinking, dynamics, and hydration in most fungal species. In addition, *R. oryzae* exhibits a noteworthy presence of fucose (Fuc; 20%)[18], glucuronic acid (GlcA; 3–23%, depending on the study)[18,19], and galactose (Gal; 12%)[18].

The atypical polysaccharide composition of Mucorales also implies a unique organization of their cell walls, but the structure of this heterogeneous biopolymer assembly has not yet been determined. The introduction of cellular solid-state NMR (ssNMR) spectroscopy has provided solutions[20,21]. Atomic-level analyses of biomacromolecules within intact fungal cells have uncovered the reorganization process of *Aspergillus* cell walls[22–24], the binding of micro-nutrients in *Schizophyllum* mycelia[25–27], and the melaninization process in *Cryptococcus* species[28–30]. *Aspergillus* cell walls are highly dynamic, undergoing significant rearrangement under stresses such as nutrient deficiency, antifungal exposure, and hypersaline conditions[31,32]. For example, analyses of *A. fumigatus* mycelia have highlighted the vital role of α−1,3-glucans in stabilizing the cell wall inner core[22,23], often overlooked by conventional chemical assays[33]. This core, supported by a pliable matrix of β-glucans, is covered by a dynamic surface shell of galactosaminogalactan, galactomannan, and proteins. Echinocandin treatment of *A. fumigatus* mycelia increases chitin, chitosan, and α−1,3-glucan levels, which interact to enhance cell wall stiffness and restrict permeability, while residual highly branched β-glucans and newly formed α−1,3-glucan structures aid in regenerating the soft matrix[34]. When *A. fumigatus* conidia shift from dormancy to growth, their α- to β-glucan ratio increases with glucans relocating to the surface and chitin embedding deeper within the inner core, while overall chitin levels remain unchanged[24].

Here, we employ ¹³C, ¹⁵N, and ¹H ssNMR to analyze five *Rhizopus* and *Mucor* species responsible for mucormycosis and CAM, with a focus on the major pathogen *Rhizopus delemar*. We observed a unique cell wall structure wherein various conformers of chitin and chitosan independently form the inner rigid layer, unsupported by any soft matrix. Only a small amount of β-glucan is present, which is associated with a specific chitin/chitosan structure to create a carbohydrate complex removable by the unique chitin synthase inhibitor nikkomycin Z. The mobile layer contains minimally branched mannan, galactan-based polysaccharides, proteins, and fucose polymers. Similar features are observed in other Mucorales species, although they differed in chitin/chitosan conformations and lacked β-glucan in the rigid phase. The structural insight explains the limited effectiveness of β-glucan inhibitors and emphasizes the potential of novel antifungal strategies targeting cell wall proteins, chitin synthesis, and deacetylation.

## Results

### Predominance of chitin and chitosan in the rigid core of *Rhizopus* cell wall

The rigid portion, crucial for the mechanical strength of the cell wall, was identified using a two-dimensional (2D) ¹³C-¹³C CORD experiment[35,36] initiated with ¹H-¹³C cross-polarization (CP), selectively detecting immobilized molecules with large ¹³C-¹H dipolar couplings (Fig. 1a). The rigid polysaccharides of *R. delemar* cell walls were very simple at the molecular level, containing predominantly chitin and its deacetylated form, chitosan (Fig. 1b). The structure of carbohydrate polymers and their distribution in dynamically distinct domains are highly reproducible (Supplementary Fig. 1). Conversely, β-1,3-glucan, a prevalent cell wall polysaccharide present in significant quantities in most fungal species, contributed minimally, representing just 5% of the rigid section as determined by intensity analysis. No other polysaccharides were found in the rigid domain of *R. delemar* cell walls. These findings were unexpected given the conventional knowledge that rigid frameworks usually consist of fibrillar macromolecules, such as plant cellulose and fungal chitin, accompanied by a significant portion of partially immobilized matrix polymers, such as pectin and hemicellulose in plants and α- and β-glucans in fungi[21]. These segments of matrix polymers in contact with fibrillar components should have been rigidified due to their interactions. The dominance of chitin and chitosan in *R. delemar*, along with minimal contributions from other polysaccharides to the rigid fraction, suggests that the chitin/chitosan core is largely spatially separated from other polysaccharides. The lack of support from a pliable glucan matrix represented a unique characteristic not identified in other cell wall systems.

At the structural level, chitin and chitosan displayed remarkable complexity, exemplified through the observation of several distinct signals for each of the C1/H1, and C4/H4 sites in the 2D hCH spectra enabled by fast magic-angle spinning (MAS) at 60 kHz (Fig. 1c), where each individual peak represents the correlations between carbon and its directly bonded proton. Proton detection in solids was challenging due to large ¹H-¹H dipolar couplings. However, these couplings are averaged out by fast MAS, enabling macromolecular structure determination through proton detection[37], including recent applications to cell walls in bacteria[38,39], plants[40,41], and fungi[26,42]. This is particularly useful for probing structural heterogeneity. For example, the ¹H chemical shift of chitin H1 (Ch1) spanned the range of 3.2–5.2 ppm, encompassing 8 distinguishable features, although one of them might originate from a weak signal of β-1,3-glucan. Notable peak multiplicity was also detected for the chitosan Cs1 and Cs4 sites, as well as the chitin Ch4 site.

Peak multiplicity was observed for ¹⁵N-¹³C cross peaks occurring between the amide nitrogen (-NH) resonating at 124 ppm and the carbon sites of chitin (Ch1, Ch4, Ch6, and CO), as well as between the amine nitrogen (-NH₂) resonating at 33 ppm and the Cs1 and Cs2 carbons of chitosan (Fig. 2a). The observed structural complexity in *R. delemar* chitin and chitosan was expected, given the inherent variations in macromolecular structures within their cellular milieu, including conformational diversity and disparities in hydrogen bonding[31,43].

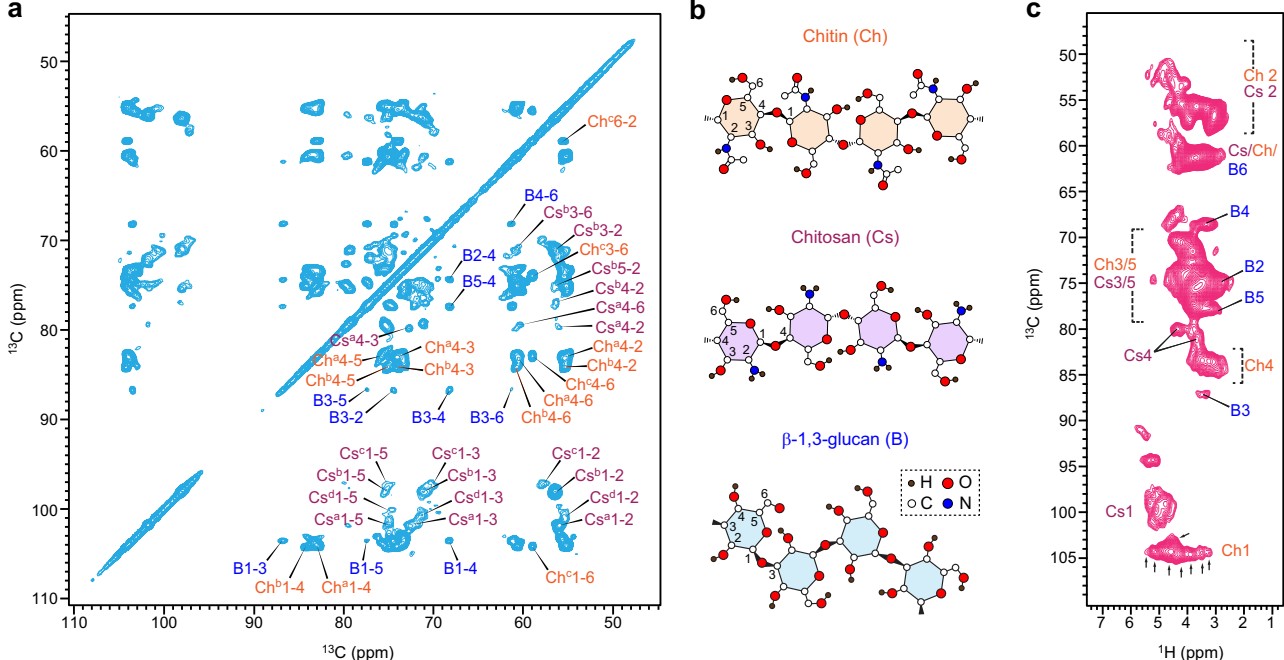

**Fig. 1 | Rigid polysaccharides in *R. delemar* cell walls. a** 2D $^{13}$C-$^{13}$C CORD spectrum selectively detecting rigid polysaccharides in the cell wall. Predominant signals were observed for chitin (Ch) and chitosan (Cs), each showing multiple sets of signals due to structural polymorphism. Superscripts indicate different subforms. For example, Ch$^a$1-4 represents the cross peaks of carbons 1 and 4 in type-a chitin. A minor set of β−1,3-glucan (B) signals was also observed. **b** Representative structure of rigid polysaccharides. **c** 2D $^{13}$C-$^1$H correlation hCH spectra showing peak multiplicity for chitin and chitosan. Arrows indicate the fine features of chitin signals. The spectra were acquired on 800 MHz NMR spectrometers. The MAS frequency is 13.5 kHz for CORD and 60 kHz for hCH.

The $^{13}$C chemical shifts of four magnetically inequivalent types of chitin forms (types a–d) were unambiguously assigned (Fig. 1a and Supplementary Figs. 2 and 3). Type-a and type-b were distinguished by their C4 chemical shifts (83.3 ppm and 84.3 ppm), whereas the distinctive C6 chemical shift of type-c at 58.9 ppm set it apart from the other types, whose C6 resonated at 60.5–60.7 ppm. When projected to 2D correlation spectra, combinations of chemical shifts from two carbons help to differentiate these chitin forms. The complete carbon connectivity of each form was shown in Supplementary Fig. 2. Three primary chitin forms were most effectively distinguished by their Ch4-6 (chitin carbon 4 to carbon 6) cross-peaks: (84.3, 60.5) ppm for chitin-a, (83.3, 60.7) ppm for chitin-b, and (83.1, 58.9) ppm for chitin-c, resulting in three spectroscopically resolved cross peaks as shown in Fig. 1a, beyond the linewidth. The minor form, type-d chitin, was mainly identified by its distinctive carbonyl and methyl sites, best resolved through its Ch1-CO, Ch3-CO, and ChMe-CO cross peaks, as shown in Supplementary Fig. 2. These spectroscopically differentiable subforms may originate from local structural changes involving hydrogen-bonds and conformational variations. Additionally, four chitosan forms were also assigned, which were better resolved, with their distinctive C1 chemical shifts (97–102 ppm) aiding in the resonance assignment.

The $^1$H and $^{15}$N signals of amide (-NH-) and amine (-NH$_2$) were resolved in 2D $^1$H-$^{15}$N correlation spectra through either heteronuclear detection (Fig. 2a; right panel) or direct proton detection (Fig. 2b). Given that the rigid portion of *R. delemar* primarily comprises cell wall polysaccharides and exhibits weaker signals of proteins, the detected amide signals should mainly originate from chitin amide, with a minor contribution from the amides of different amino acids. Similarly, the amine signals were expected to predominantly arise from chitosan, potentially accompanied by weak signals from lysine. The amine group of glucosamine (GlcN units in chitosan) could exist in either a protonated state (-NH$_3^+$) or an unprotonated state (-NH$_2$), with the former slightly favored for monosaccharide units at the pH of fungal cells. However, the exclusive resonance of amine $^1$H signals at 5.0 ppm revealed the prevalence of unprotonated amine groups, likely due to the GlcN units existing as part of a large, structured polysaccharide domain stabilized by hydrogen bonds. Otherwise, one would anticipate $^1$H peaks at approximately 8.2–8.5 ppm, as previously observed for -NH$_3^+$ in GlcN[44].

The observed sub-nanometer correlations between amide and amine groups (marked with an asterisk in Fig. 2b) suggest that a significant portion of chitin and chitosan coexist as domains within copolymers. However, there is a possibility that Lysine amine $^{15}$N resonances may contribute to this cross peak, albeit with lower intensity compared to chitosan. We performed a 2D hChH experiment using 1.67 ms $^1$H-$^1$H Radio Frequency-Driven Recoupling (RFDR)[45] to establish through-space correlations between chitin/chitosan carbon sites and chitin amide protons (H$^N$), such as between chitosan carbon 1 and chitin amide (Cs1-ChH$^N$) as shown in Fig. 2c. In the $^{13}$C chemical shift range of 10–50 ppm, no cross-peaks were observed between protein amide and protein carbons, except for the stripe of cross peaks between chitin methyl carbon and chitin protons (including ChMe-ChH$^N$). This indicates that proteins are not contributing to the long-range cross-peaks observed here, and these results confirm the existence of sub-nanometer correlations between chitin and chitosan within the same structural domain (Fig. 2d). Further discussion on $^{13}$C-based chitin-chitosan interactions will be provided in greater detail later.

**Complexity of biomacromolecules found in the mobile domain**

The mobile molecules were identified using a refocused J-INADEQUATE spectrum (Fig. 3a) initiated with a short recycle delay and $^{13}$C DP, favoring flexible molecules with fast $^{13}$C-T$_1$ relaxation. The detected molecules include proteins, lipids, and flexible carbohydrate components typically decorating the cell surface or constituting the

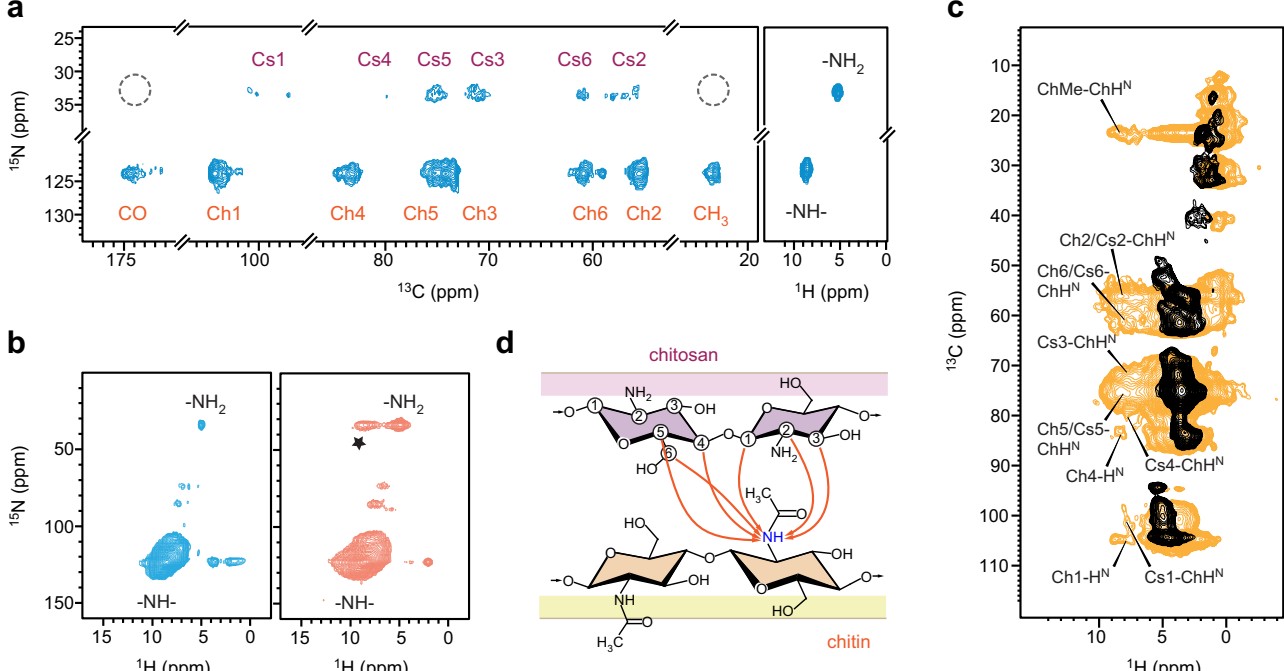

**Fig. 2 | Structural analysis of chitin and chitosan in *R. delemar*. a** 2D $^{15}$N-$^{13}$C N(CA) CX spectrum of *R. delemar* showing correlations between the carbon sites with amide (-NH-) and amine (-NH$_2$) nitrogen sites in chitin and chitosan, respectively. The absence of methyl (CH$_3$) and carbonyl (CO) motifs in chitosan is highlighted using a dashed line circle. A 2D $^{1}$H-$^{15}$N HETCOR spectrum is presented in the rigid panel, with the spectrum rotated by 90 degrees to align the amide and amine signals with the chitin and chitosan signals observed in the N(CA)CX spectrum. **b** 2D hNH spectra of *R. delemar* acquired using short (cyan; 0.2 ms) and long (orange; 2.0 ms) CP contact times. The asterisk denotes a long-range cross peak between -NH$_2$ nitrogen and -NH proton. **c** Overlay of 2D hChH the spectrum (yellow) of *R. delemar* measured with 1.67 ms RFDR mixing and a 2D hCH spectrum (black). The hChH spectrum reveals long-range $^{13}$C-$^{1}$H cross-peaks between chitin and chitosan carbons and chitin amide proton (H$^N$). **d** Illustration of chitin-chitosan interactions observed in the spectrum. All spectra of panels (**a**) and (**b**) were acquired on an 800 MHz NMR spectrometer, and spectra in panel (**c**) were measured on a 600 MHz NMR spectrometer. The MAS frequency is 60 kHz for hCH, hNH, and hChH spectra and 13.5 kHz for the $^{1}$H-$^{15}$N HETCOR spectrum.

soft matrix of the inner domain (Supplementary Fig. 4). The extensive signals of mobile polysaccharides originated from multiple sources. First, a subset of the molecules identified in the rigid portion, such as types-a and b chitosan, type-a/b chitin, and β-1,3-glucan, exhibited signals in the DP-based refocused J-INADEQUATE spectrum, designed to capture mobile components (Fig. 3b and Supplementary Fig. 5). The presence of partially mobile forms of these molecules demonstrated the creation of partially mobile sub-domains within the predominantly rigid layer.

Second, we identified the signals of α-1,2-linked and α-1,6-linked mannose residues (Fig. 3a), typically forming the polymer backbone of galactomannan. We also spotted the unique C1 signals of Gal*f* at 108 ppm (Fig. 3c), representing the sidechains galactomannan. The low intensities of Gal*f* in *R. delemar* suggest that the galactofuran side-chains in mannans are either significantly smaller or less frequent compared to those in *A. fumigatus*. Furthermore, we detected the characteristic signals of amino sugars, such as GalNAc and GalN, through comparisons with *A. fumigatus* (Fig. 3d) and with *A. nidulans* (Supplementary Fig. 6). The chemical nature of these GalNAc and GalN residues was also confirmed by their absence in a galactosaminogalactan-deficient mutant of *A. fumigatus*[22]. Alongside the observed Gal residues, GalNAc and GalN are integral components of galactosaminogalactan, a polysaccharide localized on the surface of *A. fumigatus*. Nevertheless, further investigation is needed to determine whether these sugar units in *R. delemar* form a similar structure or a different heteroglycan.

Thirdly, five distinct types of α-fucose signals were identified based on the directly bonded C5 and C6, with the latter being a methyl group displaying a unique $^{13}$C chemical shift ranging from 16 to 21 ppm

(Fig. 3e and Supplementary Fig. 7). In a recent study of *S. commune*[25], fucoses were reported to be predominantly α-1,3-linked to form polymeric structures. Linkage analysis of two Mucorales (*Phycomyces blakesleeanus* and *R. oryzae*) also suggested that fucoses are connected through α-1,3-linkage with a low degree of polymerization, resulting in their presence mainly as short segments[18]. Among the fucose residues resolved here, only type-b fucose exhibited a large, downfield C3 chemical shift at 77 ppm, characteristic of 1,3-linkages. The linkage patterns of the other four fucose units remain undetermined, but their small C1 chemical shift falling within the range of 90–98 ppm indicated α-glycosidic linkages.

Although GlcA was reported to be abundant in some *Mucor* species[18], we were unable to identify its signals in any of our spectra. To further validate this, we extracted chemical shift datasets available at the Complex Carbohydrate Magnetic Resonance Database[46] for both GlcA and galacturonic acid (GalA), which has a similar structure, to construct simulated NMR spectra[47]. Overlaying these simulated spectra with experimentally measured spectra confirmed the absence of GlcA residues in our sample (Supplementary Fig. 8). In another independent analysis of *R. delemar* grown on bread waste[19], the GlcA content was reported to be as minimal as 3%. It is conceivable that GlcA residues may be present in a loosely associated or predominantly solvated state rather than being integrated into the cell wall structure. The significance of this sugar unit's content could also fluctuate considerably based on the species considered and the specific conditions of the culture. Glucuronic acid polymers were present in high amounts as extracellular polysaccharides in the culture filtrates of Mucor mycelial cultures, but their concentration in the cell wall was dependent on the stage of development of the fungus (spore vs mycelium)[48–50].

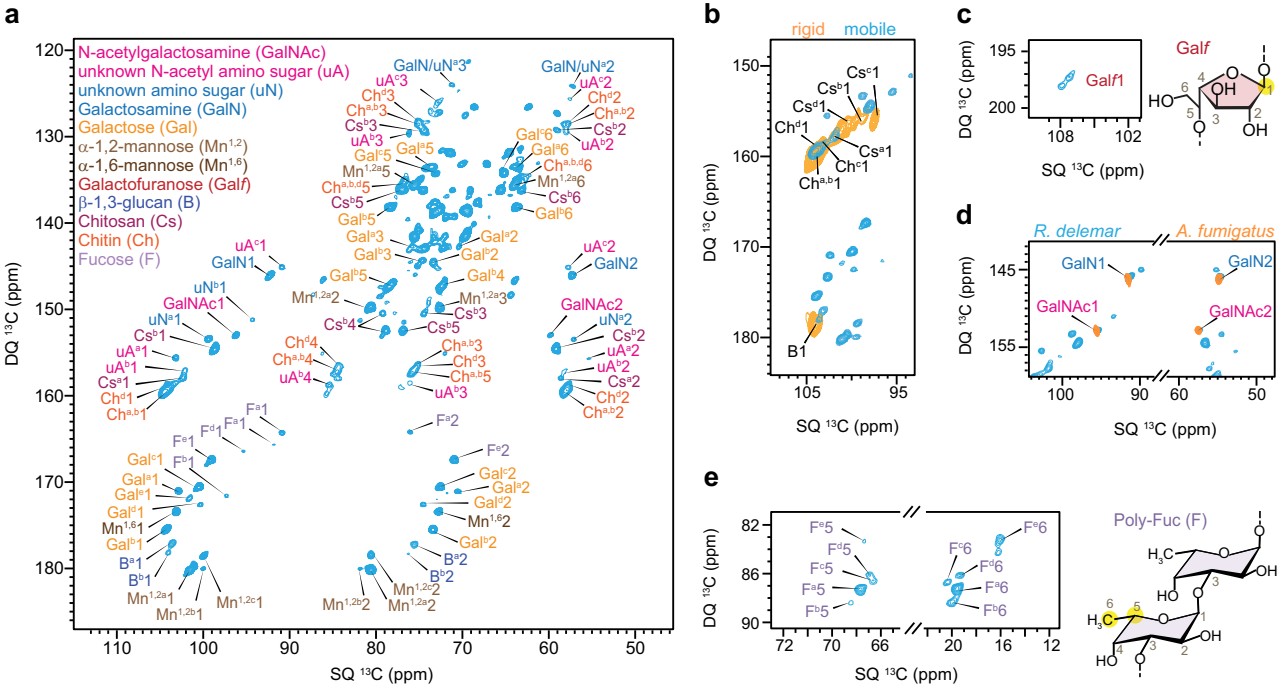

**Fig. 3 | Mobile carbohydrate components in *R. delemar*. a** 2D $^{13}$C DP refocused J-INADEQUATE spectrum of mobile carbohydrates. The signals include major monosaccharide units typically found in galactosaminogalactan (GalNAc, GalN, and Gal), GM (Gal*f*, Mn$^{1,2}$, and Mn$^{1,6}$), as well as fucose (F) residues. Some signals of chitin, chitosan, and β–1,3-glucan were also detected. Additional types of amino sugars with ambiguous chemical identities were also marked (uN and uA for amino sugars without and with acetyl groups, respectively). **b** Certain types of chitin, chitosan, and β-1,3-glucan are found in both mobile and rigid phases. The rigid and mobile molecules are detected through two J-INADEQUATE spectra measured using CP and DP as the polarization methods, respectively. **c** Weak signal of C1 (highlighted in yellow) of Gal*f* residues. The spectrum in this panel was specifically processed to show the weak signals of Gal*f*. **d** Overlay of two DP J-INADEQUATE spectra measured using *R. delemar* and *A. fumigatus* allows the identification of amino sugars in the mobile phases. **e** Unique linkage between C5 and C6 methyl carbon (shaded in yellow) allows us to resolve 5 forms of fucose residues, with at least one type with 1,3-linkages.

## Chitosan/chitin-β-glucan complex is inhibited by nikkomycin treatment

The prevalence of chitin and chitosan in the mechanically crucial core of *R. delemar* cell walls prompted us to evaluate the impact of chitin inhibitors on this fungus. We applied nikkomycin Z (Fig. 4a), the only UDP-GlcNAc analog known to block chitin synthesis in fungi[51,52], to *R. delemar* cultures. At a concentration of 16 μg/mL, *R. delemar* managed to survive, albeit with reduced growth (Fig. 4b). The growth rate decreased from 0.12 ± 0.02 mg/h for the control sample to 0.09 ± 0.01 mg/h after nikkomycin exposure, while the lag time increased from 17 ± 2 h to 22 ± 3 h. Maximal growth inhibition was 38.2%. TEM analysis revealed a nearly twofold increase in cell wall thickness, with the average rising from 119 nm to 213 nm. (Fig. 4c). These phenotypic changes caused by the addition of nikkomycin should be concomitant with molecular-level adjustments in the polysaccharide structure and composition.

Exposure to nikkomycin induced significant spectroscopic changes, notably depleting signals of linear β-1,3-glucan, type-c chitin, and type-c chitosan from the rigid domain of *R. delemar* cell walls (Fig. 4d). These molecules exhibited comparable intensities in the untreated (apo) *R. Delemar* sample, each constituting 5–6% of the rigid polysaccharides (Fig. 4e). These observations led us to hypothesize that a covalently crosslinked complex comprising these specific forms of polysaccharides was disrupted and removed by nikkomycin treatment. Although present in low quantities, this type-c chitosan/chitin-β-glucan complex is crucial for the optimal organization of *R. delemar* cell walls, as its removal significantly impacts fungal growth. Meanwhile, the spectra of mobile molecules remained largely unchanged, resembling those of the untreated sample (Supplementary Fig. 9), indicating minimal perturbation.

These findings provide new insights into the intricate process of polysaccharide biosynthesis and cell wall formation. In Ascomycetes, β-1,3-glucan and chitin synthases are transported in an inactive state to the plasma membrane, where they are assembled into activated complexes for the synthesis of insoluble polysaccharide segments[33]. These segments are then extruded into the cell wall, where the linear β-1,3-glucans undergo branching (resulting in β-1,3/1,6-glucan) and elongation, and further, form covalent crosslinks with chitin and other carbohydrates[33]. These interlinked polysaccharides constitute the fibrillar assembly that remains intact even during alkali extraction, known as the alkali-insoluble portion of the cell wall[53]. Our ssNMR data suggests that chitin-c is involved in the formation of the chitin-β-glucan complex. Nikkomycin likely inhibited the synthesis of type-c chitin, causing β-glucans to lose their crosslinking target and instead disperse into the mobile domain of the cell wall. The absence of type-c chitin as source material for deacetylation also impeded the formation of type-c chitosan.

Chitosan-c likely forms a distinct domain linked to the chitin-c domain, further crosslinking with β-1,3-glucan or β-1,3/1,6-glucan (Fig. 4f). This arrangement allows for the incorporation of the crystalline structure of chitin and the non-flat structure of chitosan, as indicated by chemical shift analysis[31], into a single cell wall diagram. At a lower occurrence, both type-c GlcNAc and GlcN may also combine to form a heteropolymer. The likelihood of chitosan-c existing as an individual polysaccharide was very low, as chitosan-c was removed simultaneously with chitin-c by nikkomycin treatment; therefore, they were expected to coexist within the same polymer.

Quantifying the fraction of the type-c chitosan-chitin-β-glucan complex in the entire cell wall is difficult because integrating compositional analyses of the mobile and rigid cell wall fractions is

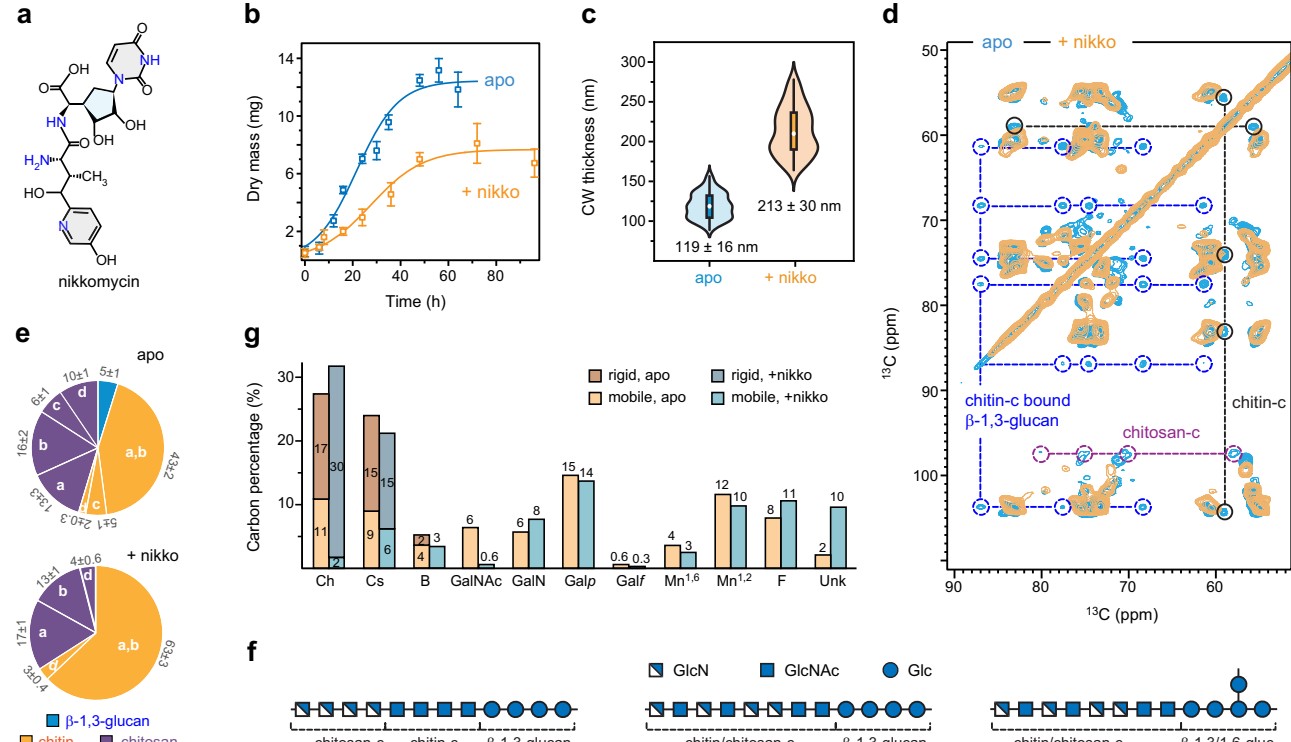

**Fig. 4 | Nikkomycin selectively removes type-c chitin/chitosan together with β-glucan. a** Chemical structure of the chitin-inhibitor nikkomycin Z. **b** Growth curve of *R. delemar* with (orange) and without (cyan) exposure to nikkomycin Z. Error bars are s.d. for triplicates. **c** Cell wall thickness measured using TEM images showing that the thickness of *R. delemar* cell wall has doubled after drug treatment. In each violet plot, the black rectangle denotes the interquartile range (IQR) from the 25th percentile to the 75th percentile, arranged in ascending order. The white circle signifies the median of the dataset, while the black vertical line indicates the standard range of 1.5 IQR. The average and standard deviation are labeled. **d** Overlay of two 2D ¹³C-¹³C 53 ms CORD spectra of apo (cyan) and nikkomycin-treated (yellow) *R. delemar* samples. The missing signals of type-c chitin (black), type-c chitosan (purple), and β−1,3-glucan (blue) signals are highlighted using circles and connected using dashed lines within each molecule. **e** Molar composition

of rigid cell wall polysaccharides estimated using peak volumes in 2D ¹³C-¹³C 53 ms CORD spectra. Error bars are s.d. **f** Illustration of the potential structures of chitosan/chitin-β-glucan complex disrupted by nikkomycin treatment generated using the Symbol Nomenclature for Glycans (SNFG). Type-c chitin and its deacetylated form, type-c chitosan, can either exist as separate domains or mix on the molecular level, which might be covalently linked to β-glucans present in either linear or branched structure. **g** Relative abundances of rigid and mobile carbohydrates in the entire fungal cell walls. The results, expressed as carbon percentages estimated from ssNMR data, are color-coded to represent the rigid and mobile fractions in both apo and nikkomycin-treated *R. delemar* samples. Brown: rigid molecules of apo sample; orange: mobile molecules of apo sample; dark blue: rigid molecules in drug-treated sample, light blue: mobile molecules of drug-treated sample. Source data are provided as a Source Data file.

challenging due to their detection via different physical principles. Mobile molecules are identified through rapid ¹³C-T₁ relaxation, enabling full magnetization recovery during short recycle delays in DP experiments. In contrast, rigid components are detected using dipolar-coupling-based ¹H-¹³C CP. To address this, we used an array of 1D ¹³C spectra (Supplementary Fig. 10) for normalization. We estimated the rigid and mobile fractions detected by ssNMR methods to be 33% and 76%, respectively, for the apo sample, with a 9% overlap, and 45% and 66%, respectively, for the nikkomycin-treated sample, with an 11% overlap (Supplementary Table 1). After depleting type-c chitin/chitosan, the levels of other chitin/chitosan forms increased, so nikkomycin treatment did not change the overall chitin/chitosan content, which remained approximately half of the carbohydrate content in both untreated and drug-treated samples (Fig. 4g and Supplementary Table 2). However, the rigid fraction of chitin and chitosan in the whole cell wall significantly increased from 32% (17% chitin and 15% chitosan) to 45% (30% chitin and 15% chitosan) following nikkomycin treatment (Fig. 4g), a critical factor for maintaining cell wall stiffness. The type-c chitosan-chitin-β-glucan complex constitutes 16% of the rigid fraction in the apo sample, accounting for only 5% of the total carbohydrate content of the whole cell. Depletion of this carbohydrate complex affects cell growth, but the restructured cell wall remains intact.

## Chitin-chitosan and carbohydrate-protein interactions stabilize the rigid core

To delve into the supramolecular assembly of the cell wall, we comparatively analyzed two 2D ¹³C-¹³C correlation spectra of apo *R. delemar* samples measured with short (53 ms CORD)[35,36] and long (1.0 s PDSD) ¹³C-¹³C mixing periods (Fig. 5a), with 1D cross sections shown in Supplementary Fig. 11. The 1.0 s PDSD spectrum unveiled 85 cross peaks that were absent in the 53 ms CORD spectrum, each of which represents an intermolecular interaction happening between two different carbohydrates at the sub-nanometer scale. These interactions mostly included chitin-chitosan (e.g., ChMe-Cs1, ChMe-Cs3, and ChMe-Cs4), chitin-β-glucan (e.g., ChMe-B5, ChCO-B5, and Chᶜ6-B5), and chitosan-β-glucan associations (such as Cs1-B5), accompanied by intra-form correlations within chitin and chitosan (e.g., Csᵇ1-Csᵃ1 and Chᶜ6-Chᵃ,ᵇMe). In addition, we identified packing interactions among different chitin forms using a 15-ms proton-assisted recoupling (PAR) spectrum (Supplementary Fig. 12)[54,55]. Chitin is the only carbohydrate that persists throughout the PAR duration, making it the most crystalline component of the cell wall. In contrast, chitosan and β-glucans, although associated with chitin, bear structural disorders and were filtered out by PAR. Meanwhile, the nikkomycin-treated sample exhibited 75 intermolecular cross peaks between different carbohydrates (Supplementary Fig. 13).

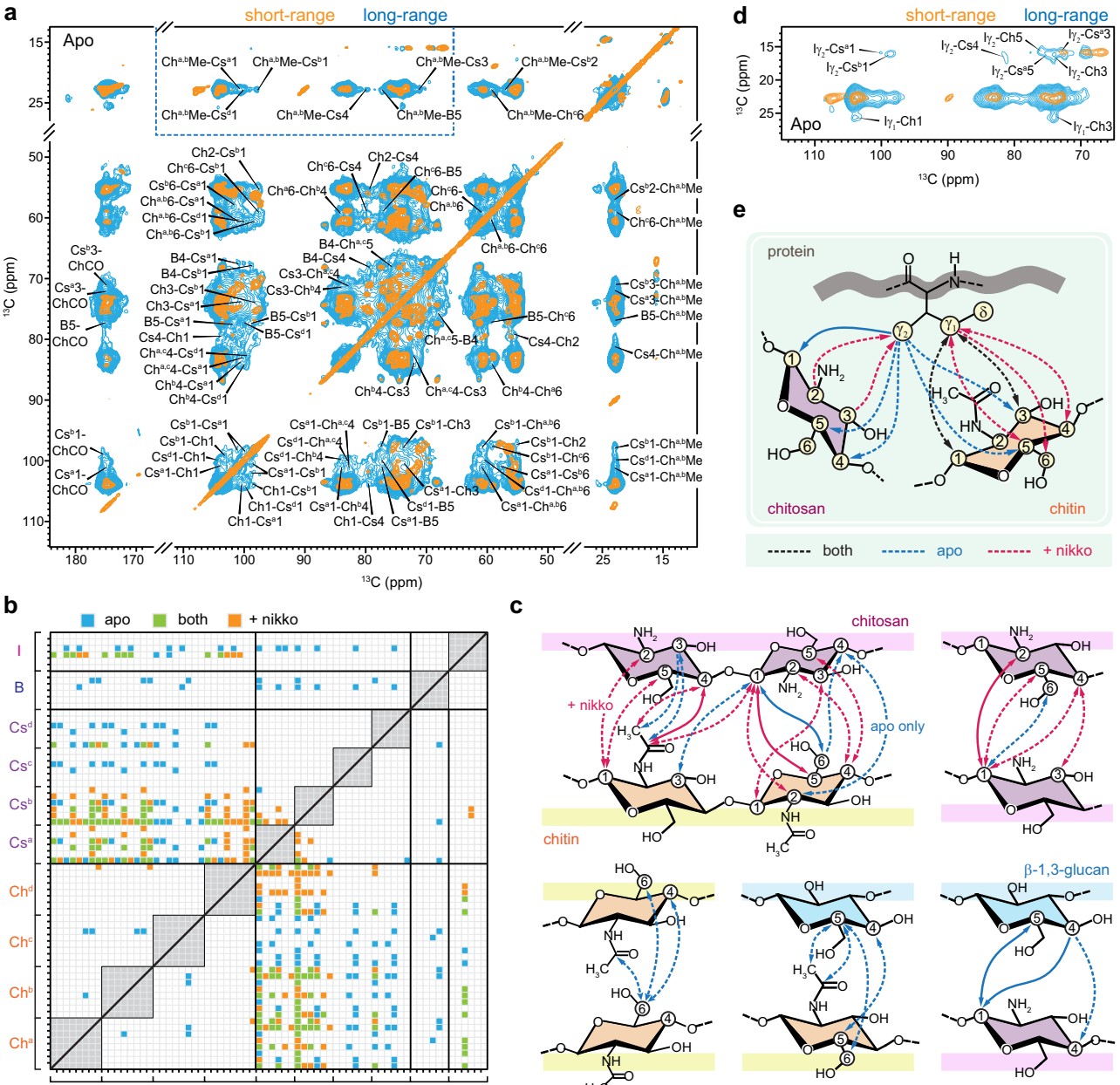

**Fig. 5 | Intermolecular interactions observed in apo and nikkomycin-treated *R. delemar*. a** Overlay of 2D ¹³C-¹³C correlation spectra with short 53-ms CORD mixing (orange) and 1-s PDSD mixing (cyan) on apo *R. delemar* sample. The dashed line box shows the spectral region where the zoomed-in view is shown in panel (**d**). **b** Spatial interactions between cell wall polysaccharides and proteins mapped onto carbon and nitrogen sites within biomolecules, including isoleucine (I), β−1,3-glucan (B), various chitosan (Cs) forms, and chitin (Ch) types. Intramolecular correlations are represented by gray boxes along the diagonal, while off-diagonal spots denote physical, intermolecular contacts observed solely in the apo sample (cyan), only in the nikkomycin-treated sample (orange), or in both samples (green). All molecular forms are depicted for interactions involving a carbon site with unspecified

allomorph assignment. **c** NMR-derived structural representation of physical contacts between cell wall polysaccharides in either the apo (blue lines) or drug-treated (magenta lines) sample, excluding those common to both. Arrowheads indicate polarization transfer directionality. Dashed lines signify the involvement of a specific subform of each of the two molecules in contact, while bold lines represent the interaction between at least two allomorphs of each molecule. Key carbon sites involved are numbered. **d** 2D ¹³C-¹³C correlation spectra of apo *R. delemar* measured with short (53 ms CORD; orange) and long (1 s PDSD; cyan) mixing showing cross-peaks between protein amino acids and carbohydrates. **e** Schematic summary of protein-carbohydrate interactions via the sidechain of an isoleucine residue.

A total of 181 intermolecular cross-peaks were observed in both apo and nikkomycin-treated *R. delemar* samples (Fig. 5b), comprising 160 inter-polysaccharide interactions and 21 carbohydrate-protein interactions. Most of these interactions occurred between chitin and chitosan, basically between the GlcNAc and GlcN residues, totaling 121 intermolecular cross peaks in these samples. Nikkomycin treatment led to the depletion of the type-c chitin-chitosan-β-glucan complex,

effectively removing interactions associated with these specific carbohydrate allomorphs (Chᶜ, Csᶜ, and B) as shown in both the statistical plot (Fig. 5b) and structural representation (Fig. 5c). However, the total number of interactions only experienced a marginal decrease, dropping from 85 to 75. This unexpected outcome was accompanied by the emergence of new interactions among the remaining chitosan and chitin forms, as well as between different chitosan forms within the

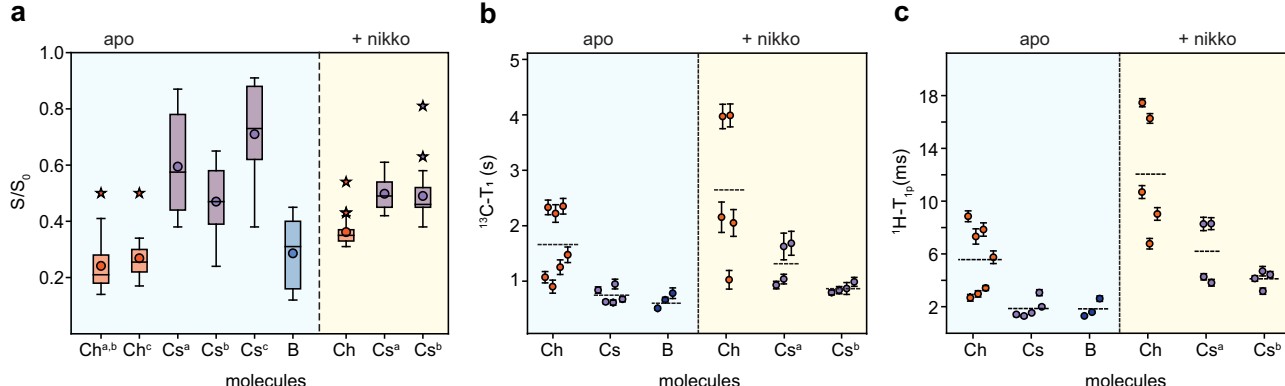

**Fig. 6 | Dynamic and hydration profile of *R. delemar* cell wall and changes by nikkomycin exposure. a** Intensity ratio ($S/S_0$) between water-edited peak intensities (S) and control spectrum ($S_0$) showing the extent of water association of cell wall polysaccharides. The x-axis lists different carbohydrates: chitin (Ch; orange), chitosan (Cs; purple), and β-1,3-glucan (B; blue), with the superscripts indicating the respective subtypes. For example, Ch^c is type-c chitin. Asterisks represent outliers, circles are average, and the horizontal straight lines are median values. The data size for the apo sample (shaded in light cyan): Ch^a,b (n = 24), Ch^c (n = 14), Cs^a (n = 18), Cs^b (n = 23), Cs^c (n = 15), B (n = 15). Data size for the nikkomycin-treated sample (shaded in light yellow): Ch^a,b (n = 24), Cs^a (n = 19), Cs^b (n = 20). **b** $^{13}C$-$T_1$ relaxation time constants of rigid cell wall polysaccharides. **c** $^1H$-$T_{1\rho}$ relaxation time constants. For both panels (**b**) and (**c**), error bars are standard deviations of the fit parameters, and horizontal dashed lines represent the average. Source data are provided as a Source Data file.

treated sample (Fig. 5b, c), which have suggested enhanced efficiency in the molecular mixing and tighter packing of the remaining chitin/chitosan domain at the nanoscale.

Although most proteins were found in the mobile phase of the cell (Supplementary Fig. 4), certain proteins within the *R. delemar* sample maintained significant structural order, which enabled them to retain their signals alongside crystalline chitin even after an extended 15 ms PAR period (Supplementary Fig. 14). Following this mixing period, most protein sites exhibited equilibrated $^{13}C$ magnetization, as evidenced by the characteristic box-shaped protein region in the spectra, indicating a molecularly homogeneous mixture within this protein domain. The carbohydrate-bound amino acid residue was identified as isoleucine predominantly in coil conformation (Fig. 5d), which was confirmed by resonance assignment of the protein spectra and secondary structure analysis (Supplementary Figs. 15, 16). Isoleucine residues account for 10–15% of the protein signals identified in the rigid fraction (Supplementary Fig. 17). Extensive interactions were observed between isoleucine carbon γ2 and various carbon sites in both chitin and chitosan, as well as between isoleucine carbon γ1 and chitin (Fig. 5d, e). This led to the hypothesis of co-localization between proteins and carbohydrates: chitosan primarily interacted with the isoleucine γ1 site, while chitin, capable of coexisting within the same chain as chitosan or existing as an individual polymer, was positioned on the opposite side and stabilized through contacts with isoleucine γ1 and γ2 sites (Fig. 5e). We did not observe cross-peaks between the isoleucine Cα/Cβ sites and carbohydrates (Supplementary Fig. 14b), likely due to these carbon sites being embedded deeper within the protein structure, with Cα potentially overlapping with chitin/chitosan C2. The absence of other amino acid residues suggests that isoleucine plays a pivotal role in maintaining the carbohydrate-protein interface.

We noticed that both type-a and type-b chitosan displayed cross peaks with isoleucine γ2, exemplified by Iγ2-Cs^a1 and Iγ2-Cs^b1, whereas type-c chitosan did not exhibit such cross peaks (Fig. 5d). Moreover, these interactions between isoleucine and chitin/chitosan were largely preserved in *R. delemar* even after nikkomycin treatment (Supplementary Fig. 13b). Taken together, these findings indicate that the interface involving isoleucine and carbohydrates is spatially distinct from the type-c chitosan-chitin-β-glucan complex depicted in Fig. 4f.

## Nikkomycin-induced decline in water retention and polymer dynamics

The hydration profiles of biopolymers were investigated by transferring water $^1H$ polarization to well-hydrated carbohydrate components (Supplementary Fig. 18)[56–58]. This water-editing experiment was carried out using a 2D $^{13}C$-$^{13}C$ format for optimal resolution (Supplementary Fig. 19). The $S/S_0$ intensity ratios between the water-edited spectrum (S) and the control spectrum ($S_0$) reflect the extent of water retention around different carbon sites (Fig. 6a). Surprisingly, chitosan exhibited the highest hydration levels in the apo *R. delemar* cell wall, with average $S/S_0$ ratios ranging from 0.47 to 0.71 across the three major subtypes. In contrast, both chitin and β-1,3-glucan displayed poor hydration, with low averaged $S/S_0$ ratios of 0.24 for the former and 0.29 for the latter. Amino acid signals identified in the rigid fraction of *R. delemar* cell walls also showed comparably low $S/S_0$ ratios of 0.1–0.3 (Supplementary Fig. 17b), indicating their low water accessibility. This suggests that these rigid proteins are likely structurally integrated into the rigid chitin/chitosan domains. Simultaneously, within the apo sample, chitin demonstrated the longest $^{13}C$-$T_1$ (Fig. 6b and Supplementary Fig. 20) and $^1H$-$T_{1\rho}$ relaxation time constants (Fig. 6c and Supplementary Fig. 21), indicating its rigidity on both nanosecond and microsecond timescales, likely due to confinement within a stiff domain. Chitosan and β-1,3-glucan exhibited similarly short $^{13}C$-$T_1$ and $^1H$-$T_{1\rho}$ relaxation time constants, revealing their relative mobility in the apo *R. delemar* cell wall.

This dataset focuses on those relatively rigid molecules essential for cell wall stiffness, which constitute one-third to half of the cell wall composition (Supplementary Table 1). In fungi, these rigid molecules refer to the partially crystalline domains of chitin/chitosan and those molecules physically stacked with them, resulting in restricted flexibility and water accessibility[59,60]. The hydrated and flexible matrix molecules account for a larger fraction of the cell wall; therefore, the cell wall is mostly dynamic.

Chitosan plays a major role in maintaining cell wall dynamics and hydration. This observed trend aligns with the inherent chemical structures of these carbohydrates. The presence of an amine group in chitosan appears to be a pivotal factor contributing to its distinct ability to associate with water, as observed in Fig. 6a. This suggests that chitosan, or the GlcN residues, should bear partial disorder to efficiently engage with water molecules. This structural concept is

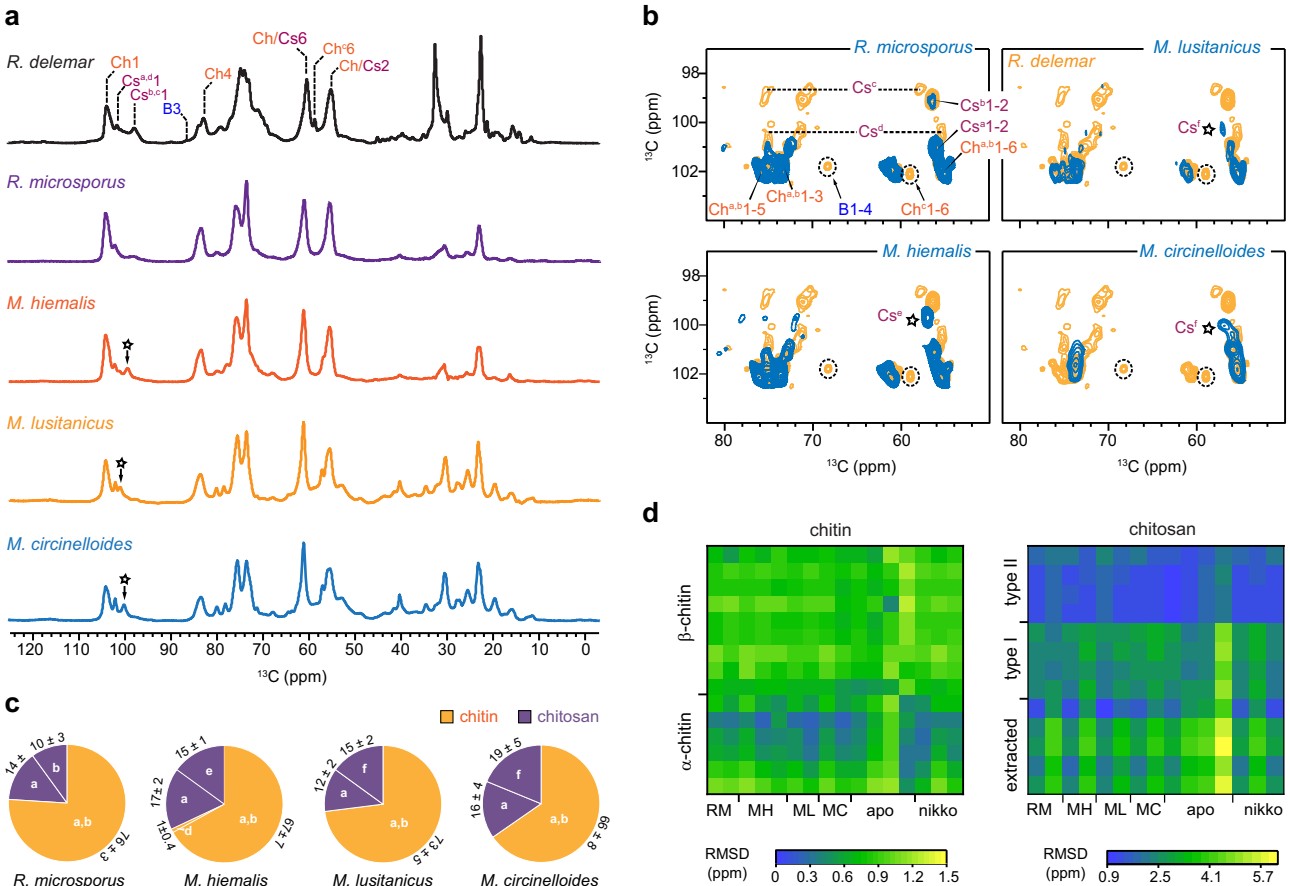

**Fig. 7 | Prevalence of chitin and chitosan across *Rhizopus* and *Mucor* species.**
**a** 1D $^{13}$C CP spectra of two *Rhizopus* and three Mucor species. Key resonance assignments are shown for resolved peaks. The sharp signals of B3 and Ch6 in *R. delemar* are absent in other species, but the majority of chitin and chitosan signals are retained. Some new chitosan signals are also spotted as marked with asterisks. **b** Rigid cell wall polysaccharides in *R. delemar* (orange) and four other fungal species responsible for CAM. Dashed lines show the types-b and d chitosan signals, while dashed circles show the signals of β–1,3-glucan and type-c chitin; these molecules are mostly absent in other species. The major type-a chitosan and type-a and b chitin are preserved in all species, but some lowly populated chitosan forms with slightly modified structures are also noted in mucor species (asterisks).

**c** Estimation of molar percentages of rigid chitin (orange) and chitosan (purple) forms in *Rhizopus* and *Mucor* species. All spectra were measured on 800 MHz NMR spectrometers at 290 K. The MAS frequency was 13.5 kHz MAS for apo *R. delemar* and 15 kHz for nikkomycin-treated *R. delemar* as well as *R. microspores* and *Mucor* samples. **d** Root mean square deviation (RMSD) heatmap of chemical shifts between model allomorphs with chitin (left) and chitosan (right) identified in *Rhizopus* and *Mucor* species. Abbreviations: *R. microsporus* (RM), *M. hiemalis* (MH), *M. lusitanicus* (ML), *M. circinelloides* (MC), and two *R. delemar* samples without (apo) and with (nikko) treatment by nikkomycin Z. Source data are provided as a Source Data file.

corroborated by the rapid relaxation behaviors and dynamic nature of chitosan (Fig. 6b, c). Considering that chitin and chitosan can exist either as individual chains or copolymers, it is inferred that chitosan is present as disordered and hydrated regions within the largely crystalline domains of chitin-chitosan and may manifest as self-aggregated subdomains or be dispersed within the structure.

The observation of β-1,3-glucan being dynamic yet poorly hydrated is also novel. In the mycelia of various *Aspergillus* species, such as *A. fumigatus* and *A. sydowii*, β-1,3-glucan is commonly a significant component forming a soft and hydrated matrix within mycelia[22,32]. The absence of water association for this molecule in *R. delemar* could potentially be linked to its minimal content and its entrapment within a crystalline inner core of the *R. delemar* cell wall.

The treatment with nikkomycin and the removal of the type-c chitin/chitosan-β-glucan complex induced perturbations in the cell wall structure, leading to a more homogenized hydration profile. Specifically, types-a and -b chitin became better hydrated, whereas type-a chitosan exhibited reduced hydration levels compared to those in the apo sample (Fig. 6a). The more uniform hydration profiles of chitin and chitosan in the nikkomycin-treated sample also indicates improved molecular mixing, enabling all molecules to be situated in a

similarly hydrated environment. This elucidates the comparable number of chitin-chitosan interactions observed when the two types of chitin/chitosan were eliminated from the cell wall by nikkomycin. Both chitin and chitosan became more rigid compared to their states in the apo samples (Fig. 6b, c). Consequently, nikkomycin induced a cell wall that is uniformly hydrated, exhibiting reduced molecular complexity but tightly packed to confer higher rigidity. This rigidification effect of the cell wall appears to be a universal mechanism employed by the fungus in response to various internal and external stressors, including carbohydrate deficiency, echinocandins, and hypersaline conditions as observed recently in different *Aspergillus* species[22,32,34] and demonstrated here in *Rhizopus* species.

### Conservation of key structural features among Mucorales
The cell wall carbohydrate fingerprints of *Rhizopus* and *Mucor* Zygomycete species were very comparable but differed significantly from other filamentous Ascomycete and Basidiomycete fungal species, as demonstrated by the distinctiveness observed in comparison to *Aspergillus* species (Supplementary Fig. 22). The predominance of chitin and chitosan is conserved in major *Rhizopus* and *Mucor* species. Comparison of 1D $^{13}$C CP spectra showed simpler profiles in *R.*

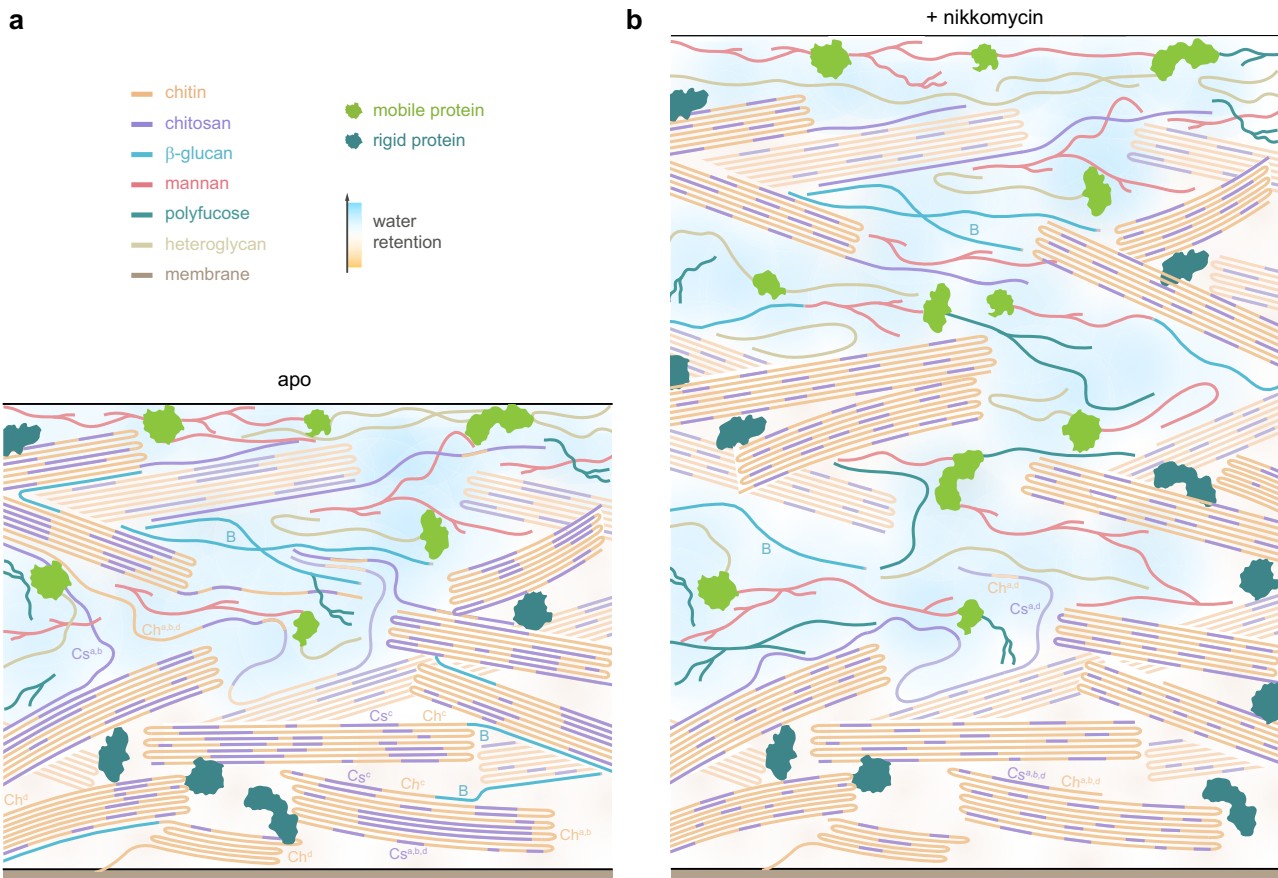

**Fig. 8 | Schematic representation of *R. delemar* cell wall organization.** Illustrative summaries are presented for (**a**) apo and (**b**) nikkomycin-treated *R. delemar* samples. Cell wall thickness has increased from approximately 120 nm to 210 nm. Chitin, organized as rigid fibrillar domains, is deposited external to the plasma membrane, featuring an antiparallel chain arrangement. These fibrils exhibit a substantial degree of deacetylation, resulting in a significant presence of chitosan, which plays a vital role in maintaining the hydration and dynamics of the rigid portion of the cell wall. Some β-glucans (B; cyan) are covalently bonded to chitin and chitosan, especially their type-c allomorphs (Ch[c] and Cs[c]). This polysaccharide complex will be depleted by nikkomycin treatment. Chitin, chitosan, and β-glucans also contribute to the soft and hydrated matrix, which also includes polymers such as mannan, polyfucose, and heteroglycans containing Gal and its amino sugar derivatives. Molecular fractions are considered, although the depicted scheme may not be strictly to scale. The background is colored to show the water gradient within the cell wall.

*microsporus*, *M. hiemalis*, *M. lusitanicus*, and *M. circinelloides* (Fig. 7a), mainly displaying a subset of polysaccharides observed in *R. delemar*. *R. microsporus* and *Mucor* species lack β-glucan in their rigid portion of cell walls (Fig. 7a, b). Chitin types a and b, along with chitosan type-a, were found to be possessed by all these fungal species (Fig. 7b, c). *R. microsporus* also exhibited type-d chitosan signals, while *M. hiemalis* displayed a chitosan signal akin to *R. delemar*'s type-d. *M. circinelloides* also exhibited some type-d chitosan signals, together with a new form established. Notably, the hypothetical carbohydrate complex involving type-c chitin/chitosan and β-1,3-glucan is absent in all *Mucor* species, as well as in *R. microsporus*, providing additional support to the proposed covalent linkage of this polymer complex. The data also suggested that the nikkomycin-treated *R. delemar* cell wall is more akin to the *Mucor* cell walls exhibiting reduced structural polymorphism. The chemical shifts of chitin and chitosan identified in these *Rhizopus* and *Mucor* species were documented in Supplementary Table 12. In addition, the mobile carbohydrates showed consistent spectral patterns across these *Rhizopus* and *Mucor* species (Supplementary Fig. 23).

Although it is challenging to reconcile the observations made on cellular samples with the structural models built on purified model materials, recent analyses of [13]C chemical shifts have shown that native fungal chitin partially resembles the α-type model with antiparallel chain packing, and native fungal chitosan resembles the non-flat helix

structure previously reported for type-II chitosan[31,61–63]. These structural characteristics are also appliable to those present in the cell walls of *Rhizopus* (*R. delemar* and *R. microsporus*) and *Mucor* (*M. hiemalis*, *M. lusitanicus*, and *M. circinelloides*) species investigated in this study, as indicated by lower chemical shift RMSDs when compared to α-chitin and type-II chitosan (Fig. 7d and Supplementary Fig. 24)[31]. This suggests that, at the molecular level, chitin and chitosan are not uniformly blended but rather form separate local clusters to maintain distinct structural attributes, whether as copolymers or individual entities.

## Discussion

Although the uniqueness of the Zygomycete cell wall has been recognized since the seminal work of Bartnicki-Garcia[64], the current high-resolution ssNMR analysis of *R. delemar* provides detailed insights into the unique structural organization of the cell wall of this species (Fig. 8a) and explained some of the responses to cell wall inhibitors. Compared to other fungi, the amount of β-1,3-glucan is extremely low in the *Rhizopus* cell wall despite that up to 3 *fks* genes (responsible for β-1,3-glucan synthesis) have been described in Zygomycetes[65]. This explains the lack of efficacy of echinocandins, which are inhibitors of β-1,3-glucan synthesis. Moreover, in agreement with the lack of structural glucan in *Rhizopus*, our study has shown that some species of Zygomycetes do not have β-1,3-glucan in the rigid domains of their cell walls, or possess it in only insignificant amounts.

From a structural standpoint, types a and b of chitin and chitosan are consistently found across *Rhizopus* and *Mucor* species (Fig. 7b) and represent the most promising targets for the development of new cell wall inhibitors. However, before proceeding, a deeper understanding of the diverse families of chitin synthases is imperative. *Rhizopus* has 23 chitin synthase (CHS) genes belonging to different classes (I/II, IV, and V/VII)[66]. Establishing connections between the specific functions of these synthase families[67,68] and the various chitin forms and structures identified by ssNMR[31] requires subsequent studies. This is especially important because chitin synthases belonging to the same family, based on sequence homologies, can have different roles in the synthesis of chitin in the cell wall[69,70].

Moreover, nikkomycin, which is a substrate analog of UDP-GlcNAc and blocks the synthetic activity of all chitin synthases tested biochemically in vitro to date, inhibits very differently the different chitin synthase proteins in different fungal species. The use of mutants has suggested that CHS1 and 2 of class I and II of Saccharomyces cerevisiae are inhibited by nikkomycin[71]. In contrast, in *A. fumigatus*, nikkomycin seems to affect only classes III, V, and VII chitin synthases, as ΔchsG, ΔCsmA, and ΔCsmB associated mutants become resistant to nikkomycin[70]. The search for chitin synthase inhibitors is further complicated in *Rhizopus* since 20 out of the 23 CHS identified in the *Rhizopus* genome are transcribed[66]. In *Rhizopus*, exposure to nikkomycin inhibits only the biosynthesis of type-c chitin. Consequently, the chitin-β-glucan complex is no longer found in the rigid core (Fig. 8b), confirming that targeting β-glucan synthesis is inappropriate. Although β-glucans and their covalently linked carbohydrate complexes are crucial for optimal fungal growth, restructured cell walls lacking these complexes still support up to half of the growth capacity in *R. delemar*. In other fungi with high amounts of β-1,3-glucans, it remains unclear which form of chitin is inhibited by nikkomycin.

In Zygomycetes, another major structural polysaccharide is chitosan, comprising a quarter of the entire cell wall (Supplementary Table 2). This differs from other fungal species where chitosan is present in low amounts and does not play an essential role in cell wall construction; multiple chitin deacetylase mutants in these fungi grow similarly to their parental strains[72,73]. Cell wall localization is predicted for 14 out of the 34 identified chitin deacetylase (CDA) genes based on potential glycosylphosphatidylinositol (GPI)-modification sites[66]. This is a significant difference from the chitin deacetylases of *Aspergillus* and *Cryptococcus*, which are either secreted or intracellular and are not GPI-anchored. The high number of membrane-associated CDAs in *Rhizopus* may contribute to its sensitivity to the new antifungal drug fosmanogepix, which blocks the GWT1 protein required for acylation of inositols during the GPI synthetic pathway[74,75]. Recently, it has been suggested to use a new class of antifungal named fosmanogepix[76] as the first-line agent against CAM[8]. However, the impact of fosmanogepix on the activity of chitin deacetylase and the reduction of chitosan content in the *Rhizopus* cell wall has not been investigated yet. Moreover, this drug may inhibit several additional general metabolic pathways other than cell wall biosynthesis, as GPI-proteins in eukaryotes serve diverse roles, including acting as receptors, adhesion molecules, enzymes, transcytotic receptors, transporters, and protease inhibitors[77].

The presence of chitosan within dynamically distinct domains might contribute to the formation of semi-dynamic regions within the rigid core, potentially mitigating the absence or low abundance of β-glucans. This echoes the recent findings on *A. sydowii*, where the role of chitosan became more prominent in the absence of α-glucan[32], another polysaccharide with dynamical heterogeneity and diverse structural functions. The ssNMR data have successfully pinpointed the polymorphic structures of chitin and chitosan, which merit further exploration for developing cell-wall-targeting antifungals. Regrettably, most currently available chitin synthesis inhibitors, such as benzoylphenylureas, are primarily insecticides and have poor antifungal activity[78,79], and only a limited number of substrate analogs are accessible for antifungal treatments, with a restricted spectrum of efficacy. Moreover, investigating the roles of chitin synthases and deacetylases in Zygomycetes genomically presents difficulties due to the large number of genes and challenges associated with constructing mutants[80].

This ssNMR study has also characterized the surface polysaccharides and opened new research avenues to understand the pathobiology of Zygomycete. Virulence factors, such as the iron uptake system or the CotH gene family, are just beginning to be analyzed in Zygomycetes[81]. The presence of homologs of galactosaminogalactan and galactomannan molecules, which have been shown to play a major role in the infectivity of *A. fumigatus*, will advocate for a better understanding of the function of these polysaccharides during infection. In addition, the role of fucosylated polymers in medical mycology remains unexplored and may provide new insights into the growth of *Rhizopus* in vivo[82].

Although type-c chitosan/chitin-β-glucan, constituting only 5% of the carbohydrate content in the apo sample, was depleted from the rigid fraction, nikkomycin exposure did not significantly alter the molecular composition (Fig. 4g). Chitin and chitosan together still makeup about half of the carbohydrates, with the degree of deacetylation remaining at 46-47%. However, their distribution and functional groups changed significantly. In the apo sample, both rigid and mobile fractions had 46% deacetylation, but post-nikkomycin, these values shifted to 75% and 33%, respectively (Fig. 8b). The mobile fraction forms also changed: from $Ch^{a,b,d}$, and $Cs^{a,b}$ in the apo sample to $Ch^{a,b}$ and $Cs^{a,d}$ after treatment. $Cs^d$, initially separated from chitosan in the apo sample (Fig. 8a), became well-mixed with chitosan after treatment (Fig. 8b). Indeed, nikkomycin led to better nanoscale mixing of chitin and chitosan, indicated by their more extensive intermolecular cross peaks (Fig. 5b), likely due to improved chitosan dispersion within crystalline domains as presented in Fig. 8b. This also explained the more uniform hydration profiles of chitin and chitosan molecules observed in nikkomycin-treated *R. delemar* (Fig. 6a). The mobile fraction, composed of β-glucan, polyfucose, Gal-based heteroglycan, and mannan, along with some disordered segments of chitin/chitosan, experienced only minor changes in relative abundance. This allowed the cell wall to maintain a functional soft matrix supporting the stiff cores (Fig. 8b).

The discovery of the isoleucine-chitin/chitosan interface in *R. delemar* is novel as this is the first direct observation of polysaccharide-protein interactions within intact fungal cells, achieved without the use of chemical extractions that might disturb native structure. Particularly noteworthy is the detection of these interactions at a thermocouple-reported temperature of 273 K, which, accounting for the heating effect of MAS, brings the sample temperature close to ambient conditions. These observations made in intact cells under physiologically relevant conditions provide compelling evidence that some protein components are integrated into or bind to the highly rigid chitin/chitosan domains, a configuration unaffected by nikkomycin treatment. This structural feature has not been observed in other fungal species[22,32], likely due to the significantly lower chitin/chitosan content in their cell walls.

The association of proteins with polysaccharides inside cell walls has been repeatedly mentioned in fungi, plants, and arthropods, but the specific nature of these interactions, whether ionic or covalent, is not always clearly defined. Many chitin-binding proteins or lectins, as well as chitinases and chitin deacetylases, could be part of the protein population associated with the chitin-chitosan scaffold. Other proteins could also have a structural function, as shown for insect cuticles, where α-chitin crystalline domains with antiparallel chain packing are attached to structural proteins containing chitin-binding domains[83,84]. This study has confirmed that ssNMR can serve as a Supplementary method for monitoring the association of polysaccharides and proteins in situ in living fungi.

This is not the first instance of observing the role of proteins, particularly their hydrophobic amino acid residues, in contributing to fungal cell wall architecture. Protein-carbohydrate covalent linkages were recently proposed for *A. fumigatus*, where valine residues were found in the alkali-insoluble portion of the cell wall, which contains covalently crosslinked polymers resisting hot alkali extraction[22]. The same valine signals were also identified in the rigid phase of the whole cell sample and vanished simultaneously with the depletion of galactosaminogalactan and galactomannan in their respective carbohydrate-deficient mutant strains[22]. These pieces of evidence indicated that structural proteins in *A. fumigatus* are attached to galactomannan and galactosaminogalactan on the surface layer, facilitated by specific valine residues[85].

The *Rhizopus* cell wall is substantially thickened and rigidified after exposure to nikkomycin, a commonly adopted strategy among fungi to shield themselves from external stressors. Similar occurrences have also been recently noted in *A. fumigatus* when combating antifungal treatment with caspofungin drugs[34], and in *A. sydowii* while surviving in hypersaline environments unsuitable for microbial growth[32]. These data suggest that the inhibitor may affect drug permeability, a concept also emerging from this ssNMR study, prompting further studies in this area. Holistically integrating biophysical insights into cell walls with efforts to elucidate the functions of diverse fungal glycoside hydrolases and glycosyltransferases and establishing connections to the polymorphic structure and versatile biological functions of polymers within the cell wall will be a necessary but challenging endeavor to fully understand Zygomycetes cell wall biosynthesis. This study on *Rhizopus* and *Mucor* species also signifies a timely expansion of ongoing efforts utilizing ssNMR techniques to comprehend fungal cell wall structures, which carry broader implications, ranging from nutrient storage[26,27] and high-performance biomaterial manufacturing[86] to fermentation and brewing[87], environmental adaptation[32], and the restructuring process relevant to invasive diseases[22,24,88].

## Methods

### Preparation of ¹³C, ¹⁵N-fungal materials

The fungal species examined in this study included *R. delemar* (FGSC-9543), *R. microsporus* (ATCC 52807), *M. hiemalis* (ATCC 20020), *M. Lusitanicus* (1006PhL), and *M. circinelloides* (CBS277.49). The procedures for preparing fungal materials for ssNMR characterization are summarized in Supplementary Fig. 25, and the catalog numbers and company names of key commercial reagents are provided below for reproducibility. Fungal conidia were cultivated at 30 °C for one week on the agar media prepared with 65 g/L YPD (Catalog # DF0428175, Thermo Fisher Scientific) and pre-autoclaved. Using inoculation loops (22-363-595, Thermo Fisher Scientific) within a sterilized laminar flow cabinet (13-261-332, Thermo Fisher Scientific), fungal conidia were subsequently transferred to a 250-mL Erlenmeyer flask containing 100 mL of pre-autoclaved ¹³C, ¹⁵N-labeled liquid growth medium. This growth medium contained Yeast Nitrogen Base without amino acids and ammonium sulfate (YNB; DF0335159, Thermo Fisher Scientific), 10.0 g/L ¹³C-glucose (CLM-1396-PK, Cambridge Isotope Laboratories) and 6.0 g/L ¹⁵N-(NH₄)₂SO₄ (NLM-713-PK, Cambridge Isotope Laboratories), with the pH adjusted to 7.0. The cultures were kept at 37 °C for 3 days in a shaking incubator (6753, Corning, LSE) with constant shaking at 200 rpm.

A parallel batch of *R. delemar* was prepared under the same conditions but with the addition of nikkomycin Z (59456-70-1, Sigma-Aldrich) at a concentration of 16 μg/mL. After cultivation, the samples were centrifuged at $7000 \times g$ for 20 min (75016014, Thermo Fisher Scientific). The resulting pellet was thoroughly washed with 10 mM phosphate buffer (PBS, pH 7.4, P3999500, Thermo Fisher Scientific) to remove water-soluble small molecules and excess ions. The harvested pellets retained the natural hydration of intact cells. Approximately 50 mg materials were packed into 3.2 mm magic-angle spinning (MAS) rotors (HZ16916, Cortecnet), and 5 mg were packed into 1.3 mm MAS rotors (HZ14752, Cortecnet) for ssNMR characterization.

### TEM analysis of cell wall thickness and morphology

TEM imaging was conducted on two *R. delemar* samples (apo and nikkomycin-treated) using a JEOL JEM-1400 electron microscope in the MSU Center for Advanced Microscopy. The fungal mycelia were fixed with 2.5% glutaraldehyde and 2% paraformaldehyde (15700, Electron Microscopy Sciences) in 0.1 M cacodylate buffer, followed by embedding in 2% agarose and post-fixed in 0.1 M osmium tetroxide (SKU 19152, Electron Microscopy Science). Dehydration was achieved through a series of acetone solutions with increasing concentrations, followed by infiltration with epoxy resins and acetone in proportions of 25:75, 50:50, and 75:25, respectively. The samples were left in the 75:25 epoxy resin and acetone solution overnight, then treated with 100% resin for two days, with multiple resin changes. Finally, the sample was placed in an oven at 70 °C to prepare the blocks. Ultrathin sections were cut using a LEICA EM UC7 microtome, stained with 1% uranyl acetate (22400, Electron Microscopy Sciences) and lead acetate (17800, Electron Microscopy Sciences), and mounted on carbon-coated grids (FCF-150-CU, Electron Microscopy Sciences). TEM imaging focused on cross-sections of the hyphae, with 100 measurements of the cell wall performed for each group (Supplementary Table 3). Cell wall thickness was measured using ImageJ (software version V1.8.0_172), and data were analyzed statistically with the $t$ test.

### Growth profile of apo and nikkomycin-treated *R. delemar* cultures

One-week-old conidia were collected and suspended in 0.01% Tween-20 (97062-332, VWR). The suspension was adjusted to a final concentration of $2 \times 10^6$ cells/mL (Supplementary Fig. 26). Flasks were incubated at 30 °C with a shaking rate of 200 rpm. At designated intervals, pump filtration was performed to determine fungal mass. The growth curve was fitted using a sigmoid equation[89]:

$$m = m0 + \frac{A}{1 + \exp(-k(t - t_{1/2}))} \tag{1}$$

where m represents the mass after a specific incubation time, k is the buildup rate and $t_{1/2}$ is the constant defining the half-time of the fungal transition from the lag phase to the maturation phase. The lag time of fungal growth can be calculated as:

$$t_{\text{lag}} = t_{1/2} - 2/k \tag{2}$$

### SsNMR resonance assignment

High-resolution 1D and 2D ssNMR experiments were measured on an 800 MHz (18.8 Tesla) Avance III spectrometer at the National High Magnetic Field Laboratory, Tallahassee, FL, and an 800 MHz Avance Neo spectrometer at Michigan State University. On both spectrometers, 3.2-mm MAS HCN probes were used with the MAS frequency set to 13.5 kHz for apo *R. delemar* and 15 kHz for nikkomycin-treated *R. delemar*, as well as *R. microspores* and *Mucor* samples. The experiments were performed at 290 K unless specifically mentioned. ¹³C chemical shifts were referenced externally by calibrating the adamantane CH₂ peak to 38.48 ppm and applying the resulting spectral reference (sr) value to fungal spectra. ¹⁵N chemical shifts were referenced externally to the liquid ammonia scale, using the methionine amide resonance (127.88 ppm) of the model tri-peptide N-formyl-Met-Leu-Phe-OH (CC10073, Cortecnet) for calibration. Unless specifically mentioned, typical radiofrequency field strengths were 83–100 kHz for ¹H decoupling, 62.5 kHz for ¹H hard pulses, 50–62.5 kHz for ¹³C, and 41 kHz for ¹⁵N. The experimental parameters of all NMR spectra were

documented in Supplementary Table 4. All NMR spectra were collected in Topspin 3.5, and spectral analysis was performed using Topspin 4.0.8.

One-dimensional (1D) $^{13}$C spectra were acquired using various polarization methods to detect molecules in different regimes of dynamics. Rigid components were detected through dipolar-based 1D $^{13}$C cross-polarization (CP) with a 1-ms contact time, adhering to Hartmann-Hahn match conditions of 62.5 kHz for $^{13}$C and $^1$H. For quantitative analysis, we coupled 1D $^{13}$C direct polarization (DP) with a long recycle delay of 30 s, ensuring complete relaxation back to equilibrium for all molecules before the subsequent scan. Shortening the recycle delay to a substantially shorter period of 2 s in the same $^{13}$C DP experiment allowed us to selectively detect the mobile molecules with fast $^{13}$C-$T_1$ relaxation. These 1D experiments were measured on three independently prepared replicates of *R. delemar* samples, which are highly reproducible (Supplementary Fig. 1).

To facilitate resonance assignment, 2D $^{13}$C-$^{13}$C 53-ms CORD homonuclear correlation spectra were recorded for all fungal samples[35,36]. These spectra showed the intramolecular cross-peaks between different carbon sites within each molecule (Supplementary Fig. 2). In addition, 2D DP $^{13}$C refocused J-INADEQUATE experiments[90,91] were conducted with either CP or DP (1.5 s recycle delay) to detect molecules in the rigid and mobile domains, respectively (Supplementary Fig. 4). This experiment correlated each double-quantum (DQ) chemical shift with two corresponding single-quantum (SQ) chemical shifts to generate asymmetric spectra that allowed us to efficiently track the through-bond carbon-connectivity within each molecule. Each τ period (out of the four) was set to 2.3 ms for optimized detection of carbohydrates on our instruments. Furthermore, 2D $^{15}$N-$^{13}$C N(CA)CX heteronuclear correlation spectra were acquired for probing signals from chitin amide and chitosan amine[92]. For this experiment, 0.6-ms $^1$H-$^{15}$N CP, 5-ms $^{15}$N-$^{13}$C CP contact times, and 100-ms DARR mixing time were used. All 1D experiments, as well as the 2D CORD experiment, measured all five *Rhizopus* and *Mucor* strains, while the INADEQUATE experiment and $^{15}$N-$^{13}$C correlation experiments were only measured on *R. delemar* samples (Supplementary Table 4). The assigned chemical shifts of rigid and mobile polysaccharides were documented in Supplementary Tables 5 and 6, respectively. The chemical shifts of amino acid residues were tabulated in Supplementary Table 7, with their secondary structure and composition presented in Supplementary Figs. 16 and 17, respectively. All NMR spectra were processed using Topspin 4.0.8. Figures were prepared using Adobe Illustrator Cs6 V16.0.0. Illustrative representations were provided for carbohydrate components using either their chemical structure or Symbol Nomenclature for Glycans (SNFG)[93].

### Compositional analysis of rigid carbohydrates

The molar percentages of rigid carbohydrates were determined by analyzing cross-peaks in CORD spectra. Peak volumes were acquired using the integration function in the Bruker Topspin software. To minimize uncertainties caused by spectral overlap, we only used well-resolved peaks for intensity analysis (Supplementary Table 8). The mean of the volumes of resolved cross peaks was then calculated following a recently established protocol[94]. The molar percentage of a specific polysaccharide was determined by normalizing the sum of peak integrals with their corresponding counts, as expressed in the following equation:

$$RA^{poly.^x}(\%) = \frac{\sum_{n=1}^{n_{peaks}^{poly.^x}} I_n^{poly.^x} / n_{peaks}^{poly.^x}}{\sum_{m=1}^{m^{poly.}} \left( \sum_{n=1}^{n_{peaks}^{poly.^x}} I_n^{poly.^x} / n_{peaks}^{poly.^x} \right)} \times 100 \quad (3)$$

Where $n_{peaks}^{poly.^x}$ represents the number of cross peaks in CORD, and $I_n^{poly.^x}$ denotes the integral of each individual peak volume.

The standard error for the polysaccharide ($SE^{poly.^x}$) was calculated by dividing the standard deviation of the peak volume by the sum of the cross-peak count. The overall standard error ($\sum SE$) was calculated by taking the square root of the sum of the squared standard error for each polysaccharide. Subsequently, the polysaccharide's percentage error ($PE^{poly.^x}$) was calculated by dividing the standard error ($SE^{poly.^x}$) by the average peak volume of the specific polysaccharide ($\bar{x}^{V^{poly.^x}}$), then multiplying it by the ratio of the standard error of the specific polysaccharide ($\sum SE$) to the total peak volume ($\sum V^{poly.}$), further multiplied by the percentage of composition of the corresponding polysaccharide ($RA^{poly.^x}$):

$$PE^{poly.^x} = \frac{SE^{poly.^x}}{\bar{x}^{V^{poly.^x}}} \times \frac{\sum SE}{\sum V^{poly.}} \times RA^{poly.^x} \quad (4)$$

### Fast-MAS ssNMR experiments for proton detection

The 2D hCH and 2D hNH spectra in proton detection were carried out under fast MAS using a Bruker Avance III 800 MHz spectrometer located at the National High Magnetic Field Laboratory (Tallahassee, Florida). A homebuilt 1.3 mm triple-resonance MAS probe was utilized, operating at a MAS frequency of 60 kHz. The $^{13}$C and $^{15}$N chemical shifts were referenced using MLF as an external reference, and a small amount of sodium trimethylsilylpropanesulfonate (DSS) was added to the sample to calibrate the $^1$H chemical shift. For each 2D hCH and 2D hNH, a total of 256 TD points were acquired, and for each TD point, 16 scans were coadded with a recycle delay of 2 s. The swept-low-power TPPM decoupling[95] was applied during the $t_1$ evolution period on the $^1$H channel, while waltz-16 decoupling[96] was applied during the $t_2$ evolution period on the $^{13}$C channel. A radiofrequency field of 10 kHz was employed for both decoupling sequences. For 2D hCH, the first HC CP contact time was optimized to 200 μs, and the second CH CP contact time was set to 200 μs. During both CP blocks, the rf field of 40 kHz was applied on the $^1$H channel while at 20 kHz was applied for the $^{13}$C channel. For 2D hNH, the first HN CP contact time was optimized to 2 ms, and the second NH CP contact time was set to 0.2 ms for connecting short-range correlations and 2.0 ms for measuring long-range correlations. The water resonances, as well as direct magnetization from protons, were suppressed using the MISSISSIPPI sequence[97] on $^1$H channel. The total experimental time was 2.3 h for each 2D. The data were acquired using the States-TPPI method[98].

The 2D hChH correlation experiment[38] was performed on a 600 MHz Bruker Avance Neo spectrometer at Michigan State University, equipped with the 1.3 mm triple resonance MAS probe, with the sample spinning at a frequency of 60 kHz. The hChH experiment utilized RFDR mixing times of 1.67 ms[45]. A total of 256 TD points were collected, with 32 transients co-added for each TD and a recycle delay of 2.5 s. The total experimental time was 5.6 hours. Concurrently, a 2D hCH spectrum was collected on the same Bruker 600 MHz spectrometer under identical conditions but without RFDR mixing for direct comparison. The 2D data were collected using the States-TPPI method[98]. The sample temperature was 302 K, which was measured based on water $^1$H chemical shift with respect to the DSS signal at 0 ppm. The experimental details were tabulated in Supplementary Table 4.

### Analysis of intermolecular packing

To identify intermolecular interactions among carbohydrate polymers and between carbohydrates and proteins, we conducted 2D $^{13}$C-$^{13}$C proton-driven spin diffusion (PDSD) experiments on both the apo and nikkomycin-treated *R. delemar* samples, employing a very long mixing time of 1.0 s. These experiments were conducted at 273 K to partially restrict the dynamics of molecules, improving the efficiency of polarization transfer, and better preserving the intensities after the long mixing time. In addition, a 15-ms proton-assisted recoupling (PAR)

experiment[54,55] was carried out on the apo *R. delemar* sample, with the field strengths of 56 kHz for $^{13}$C and 53 kHz for $^1$H during the PAR period. However, most signals did not endure the long PAR period. The remaining magnetization following PAR mixing is greatly influenced by both the PAR mixing time and the relaxation rates, which favors molecules with structural order, as demonstrated in recent studies involving amyloid fibrils[99]. The residual carbohydrate signals were only from the highly crystalline chitin (Supplementary Fig. 12), and this allowed us to cleanly detect interactions of different chitin forms. The intermolecular cross peaks are summarized in Supplementary Table 9.

## SsNMR experiments of dynamics and hydration

SsNMR experiments on polymer dynamics and water contacts were conducted on a Bruker Avance Neo 400 MHz (9.4 Tesla) spectrometer using a 3.2 mm HCN probe and on a Bruker Avance 400 MHz spectrometer using a 4.0 mm HXY probe. To evaluate the water accessibility of polysaccharides, water-edited 1D and 2D $^{13}$C-detection experiments were employed[56–58]. The experiment began with $^1$H excitation followed by a $^1$H-$T_2$ filter of 1.0 ms × 2 for apo sample and 1.4 ms × 2 for nikkomycin-treated sample, eliminating 97% of the polysaccharide signals while retaining 82% of water magnetization (Supplementary Fig. 18). Subsequently, the water $^1$H magnetization was transferred to polysaccharides through a 4-ms $^1$H mixing period and then transferred to $^{13}$C via $^1$H-$^{13}$C CP for high-resolution $^{13}$C detection. A 50-ms DARR mixing period was implemented to the water-edited spectrum, showing exclusively well-hydrated molecules, and the control 2D spectrum, displaying full intensities. The relative intensity ratios (S/$S_0$) between the water-edited spectrum (S) and the control spectrum ($S_0$) were quantified for all resolvable carbon sites to reveal the relative extent of hydration (Supplementary Table 10). The intensities were pre-normalized by the number of scans of each spectrum.

The $^{13}$C-$T_1$ relaxation was measured using the Torchia-CP experimental scheme[100] where the z-filter duration ranged from 0.1 µs to 8 s. For each resolved peak, the decay of its intensity was quantified as the z-filter time increased, and the data were fit to a single exponential equation to obtain the $^{13}$C-$T_1$ relaxation time constant (Supplementary Fig. 20). The absolute intensity of each peak was pre-normalized by the number of scans. For measuring $^{13}$C-detected $^1$H-$T_{1\rho}$ relaxation, a Lee-Goldburg (LG) spinlock sequence combined with LG-CP was employed[101,102], suppressing $^1$H spin diffusion during both the spinlock and the CP contact. This allowed site-specific measurement of the $^1$H-$T_{1\rho}$ relaxation of protons sites through the $^{13}$C-detection of their directly bonded carbons. The decay of peak intensity was fit to a single exponential function to obtain the $^1$H-$T_{1\rho}$ time constants (Supplementary Fig. 21 and Table 11). Relaxation curves were fitted using OriginPro 9.

### Reporting summary

Further information on research design is available in the Nature Portfolio Reporting Summary linked to this article.

## Data availability

All relevant data that support the findings of this study are provided in the article and supplementary Information. The original Topspin NMR datasets have been deposited in the Zenodo repository under the DOI number https://doi.org/10.5281/zenodo.13507960. Source data are provided with this paper.

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

## Acknowledgements

This work was supported by the National Institutes of Health (NIH) grants AI173270 to T.W. and AI168867 to P.W. A portion of this work was performed at the National High Magnetic Field Laboratory, which is supported by the National Science Foundation Cooperative Agreement No. DMR-2128556 and the State of Florida.

## Author contributions

Q.C. and M.C.D.W. performed $^{13}$C and $^{15}$N NMR experiments. Q.C. conducted TEM and growth assays. J.R.Y. and A.A. conducted fast

MAS [1]H-detection experiments. P.W. and Q.C. prepared fungal cultures. Q.C., M.C.D.W., J.R.Y., A.A., and J.P.L., analyzed the data. T.W. designed the experiments. All authors contributed to the writing of the manuscript.

## Competing interests

The authors declare no competing interests.
