## [Peer Review File · Nature Communications]

Molecular architecture of chitin and chitosan-dominated cell walls in Zygomycetous fungal pathogens by solid-state NMRREVIEWER COMMENTS

Reviewer #1 (Remarks to the Author):

Building on a well-developed set of solid-state NMR experiments and resonance assignments (initiated in the literature by this group of investigators for *Aspergillus fumigatus* in 2018), Cheng, et al. have extended their purview to a set of structurally intriguing Mucorales (mainly *Rhizopus delemar*) fungal species that pose significant dangers to human health. By again tailoring their experiments to focus on rigid, mobile, or hydrated cell-wall domains as well as spatial relationships of various constituent pairs, their findings demonstrate the compositional and organizational distinctiveness of cell walls in this fungal order. Nonetheless, the manuscript could be strengthened by providing or proposing additional physical data to support some of their speculations about the organization of the various chemical constituents and their reorganization in response to nikkomyacin treatments. Several examples may illustrate the findings that stood out along with some reservations regarding their interpretation.

Notable compositional trends included the dominance of rigid chitin and chitosan as compared with β -glucans (replicated in several *Rhizopus* and *Mucor* species), the conformational diversity of each abundant polysaccharide (PS) constituent, the variety of mobile molecular species, and the selective depletion of particular chitin/chitosan allomorphs and β -1,3-glucans upon treatment with the chitin synthesis inhibitor nikkomyacin. This last finding could not be made without the detailed analyses presented by the authors. Of greater potential practical significance is the hypothesis that emerges and is then confirmed from these observations: disruption of a complex among these three PS types. However, the logic of proposing that the β -glucans now lack their crosslinking partners seemed insufficiently grounded. That is, the sensitivity of Mucorales to echinocandins that inhibit β -1,3-glucan synthesis (p. 11) makes sense because those constituents are found to be sparse among the rigid components, but is it more challenging to wipe out enough of the abundant chitin and chitosan to disrupt the 3-way complex entirely? If only a subset of chitin and chitosan allomorphs is complexed, then we need to know what proportions they represent to evaluate the argument.

The dominance of chitin and chitosan in *R. delemar* also allowed the investigators to report the first observation of through-space protein-PS interactions within intact fungal cell walls, independently of whether the cells were treated with nikkomyacin. The authors suggest that the proteins are either integrated into the rigid chitin/chitosan domains or bound to the surface of these lamellar domains (p. 15), but these speculations would be more compelling if the authors could offer some means to distinguish between these options.

Finally, this report includes new observations about constituent-specific hydration based on polarization transfer NMR experiments. Hydration levels were robust for chitosan and poor for both chitin and β -1,3-glucans. The proposed explanations (p. 17) invoked the relative amounts of these constituents, but is it appropriate to consider only the rigid domains? If both rigid and mobile populations must be accounted for, then we need to know their relative proportions to make cogent arguments.

A related reservation concerns the proposal of "vast highly crystalline domains" described on p. 19 and in Figure 7; do we know what proportion of each constituent is found in experiments that favor the rigid or mobile domain, and that no constituents are double counted in both experiments? Although the authors' attempt to construct a model for this complex cell-wall system is laudable, the rigor of their reasoning was not always clear to this reader.

Reviewer #2 (Remarks to the Author):

This work uses (primarily) solid-state NMR to study cell walls in five Mucorales species. The paper greatly advances our current knowledge of the cell wall organization both in terms of molecular detail as well as regarding its structural and dynamical properties. Importantly, this information is

obtained both before and after treatment with an antifungal inhibitor. However, the authors should clarify several aspects of the research results, their analysis and how they lead towards a unified structural model.

[1] The paper presents a refined model of the *R. delemar* cell wall organization but does not depict how the cell wall organization is altered after nikkomycin treatment. Since this inhibitor significantly alters the cell wall dimensions and the paper contains highly relevant EM and extensive NMR data of such preparations, such a model should be included in the final figure. For the sake of clarity, it may also be useful to separate results and discussion sections in the text. In addition, it may be useful to comment in a new figure 7 on the location of the different Ch/Cs species that are not found in the current version of Fig. 7.

[2] The data confirm earlier reports that the rigid core of the cell wall mainly comprises chitin and chitosan, with smaller quantities of β -glucans. In addition, the authors speculate about the formation of a β -glucan-chitin/chitosan complex. To the reviewer, the experimental evidence for such an assembly needs further clarifications. Firstly, the NMR data indeed suggest polymorphism with at least 4 different spectroscopic species for chitin and chitosan (a-d). According to lines 261-268, the notion of a β -glucan-chitin/chitosan complex seems to primarily be based on the influence of nikkomycin on chitin/chitosan subspecies 'c' which only contributes a minor part to the entire chitin/chitosan pool (Fig. 3, e). If it is true that nikkomycin treatment almost doubles the cell wall thickness (Fig. 3c), it seems difficult to imagine that the disappearance of such complexes is the only difference in cell wall organization before and after treatment (see comment 1). Along similar lines, Page 7/ Figure 7 suggest that a significant portion of chitin and chitosan exist as domains within copolymers. Notably, the NHN experiment (fig. 1e) contains only one rather weak cross peak and uses long mixing times that may compromise a detailed structural analysis. In addition, Lysine (observed in the amino-acid analysis) side-chain signals are found at similar ^{15}N chemical-shift values. The claim (lines 186-188) that "The observed sub-nanometer correlations between amide and amine groups ... provide additional evidence supporting the idea that a significant portion of chitin and chitosan exist as domains within copolymers." needs further clarification.

[3] To probe the supramolecular assembly, the authors mainly use long NMR mixing times. The spectrum (Figure 4a) contains mostly broad correlations that are often difficult to attribute to individual peaks seen in the "short-range" experiment. Even in spectral regions that only contain one or a limited set of "short-range" correlations (e.g. around 78 ppm, 100 ppm, 25 ppm) broad, featureless peaks are found. It would help if the authors (possibly in the SI) exemplify some of the additional correlations using vertical/horizontal dashed lines in the spectrum. In addition, they report in the SI (SI Figure 10 & caption) that a short (15 ms) PAR spectrum only retained chitin signal, "revealing the disordered nature of β -glucan and chitosan signals..." If this is the case, would the latter observations not speak against the co-existence of chitin-chitosan fibers as suggested in the model of Figure 7?

[4] The authors manage to identify a number of amino-acid signals and report chemical-shift assignments (table S5). In addition, they should include an analysis of these assignments in terms of protein secondary structure for both (rigid & mobile) protein species. Also, further information about the relative abundance of the different amino acids would be helpful. Can such information be correlated to polypeptides/proteins known to be associated with Mucorales cell walls? The abstract also claims that "some proteins are entrapped within this semi-crystalline chitin/chitosan layer, stabilized by the sidechains of hydrophobic amino acid residues, and situated distantly from β -glucans". This claim seems to be mainly based on Fig. 4d/e which reveals weak correlations between the amino-acid methyl carbons and Cs/Ch NMR signals. Please indicate this spectral region in Figure 4a and comment whether also Ca/Cb correlations to Ch/Cs occur. Did the authors observe similar correlations for other "rigid" amino acids? If not, please explain.

[5] Minor points:

- Lines 131-133: It is not clear what the authors mean. ^1H detection on fungal cell walls has already been demonstrated before. Please clarify or remove. Also, they write " For example, the ^1H chemical shift of chitin H1 (Ch1) spanned the range of 3.2-5.2 ppm encompassing at least 8 distinguishable features.." Please indicate those in the figure. To the reviewer, the corresponding

cross peak region is rather featureless which may indeed reflect chitin chain heterogeneity as regularly seen in MAS NMR of biopolymers. The same applies to the NC chitin/chitosan correlations.

- NMR Chemical shift signatures of alpha- and beta-chitin structure have been published and are cited in this work. To the reviewer, it would be useful to compare the current assignments to these literature values in more detail. The chemical shift variation reported for the different Ch,Cs species (table S3) is often rather small (0.1-0.3 ppm) which probably is smaller than the actual ^{13}C line width. Please clarify which of the variations are actually spectroscopically significant and whether those cannot be attributed to local changes such as ^1H -bonding or molecular packing.

Typographical error: Line 276: constitute

We appreciate the advice from both reviewers, and we have addressed all the comments in a point-to-point manner as detailed below. **Two versions of the manuscript files**, one with all changes highlighted in blue and the other with all changes tracked, are provided to facilitate the review process.

Responses to Reviewers' Comments

Reviewer #1

Building on a well-developed set of solid-state NMR experiments and resonance assignments (initiated in the literature by this group of investigators for *Aspergillus fumigatus* in 2018), Cheng, et al. have extended their purview to a set of structurally intriguing Mucorales (mainly *Rhizopus delemar*) fungal species that pose significant dangers to human health. By again tailoring their experiments to focus on rigid, mobile, or hydrated cell-wall domains as well as spatial relationships of various constituent pairs, their findings demonstrate the compositional and organizational distinctiveness of cell walls in this fungal order. Nonetheless, the manuscript could be strengthened by providing or proposing additional physical data to support some of their speculations about the organization of the various chemical constituents and their reorganization in response to nikkomycin treatments. Several examples may illustrate the findings that stood out along with some reservations regarding their interpretation.

We appreciate the positive comments regarding the technical merit and biological significance of this study. We also value the insightful advice on improving the manuscript. We have implemented all the suggested changes and provided additional datasets as detailed below.

Notable compositional trends included the dominance of rigid chitin and chitosan as compared with β -glucans (replicated in several *Rhizopus* and *Mucor* species), the conformational diversity of each abundant polysaccharide (PS) constituent, the variety of mobile molecular species, and the selective depletion of particular chitin/chitosan allomorphs and β -1,3-glucans upon treatment with the chitin synthesis inhibitor nikkomycin. This last finding could not be made without the detailed analyses presented by the authors. Of greater potential practical significance is the hypothesis that emerges and is then confirmed from these observations: disruption of a complex among these three PS types. However, the logic of proposing that the β -glucans now lack their crosslinking partners seemed insufficiently grounded. That is, the sensitivity of Mucorales to echinocandins that inhibit β -1,3-glucan synthesis (p. 11) makes sense because those constituents are found to be sparse among the rigid components, but is it more challenging to wipe out enough of the abundant chitin and chitosan to disrupt the 3-way complex entirely? If only a subset of chitin and chitosan allomorphs is complexed, then we need to know what proportions they represent to evaluate the argument.

We appreciate the constructive advice. It has been technically difficult to unify the compositional results from the rigid and mobile fractions of fungal cell walls, as they are probed using different physical principles. Here, we introduced an approach to estimate the percentage of the polysaccharide complex relative to the carbohydrate content of the whole cell sample. The type-c chitosan-chitin- β -glucan complex accounts for 5% of the carbohydrate content within the whole cell wall. The overall chitin/chitosan content relative to the whole cell wall has not changed significantly, remaining at 51-

53% in both apo and nikkomycin-treated samples. We also noted a significant increase in the presence of chitin/chitosan in the rigid fraction from 32% to 45%. The results are now described in a new paragraph in **Lines 320-337** and **Fig. 4g**. The detailed procedures are provided as new **Supplementary Tables 1** and **2** and supported by a new **Supplementary Fig. 10**. These changes help elucidate how the *Rhizopus* cell wall is modified by nikkomycin treatment.

The dominance of chitin and chitosan in *R. delemar* also allowed the investigators to report the first observation of through-space protein-PS interactions within intact fungal cell walls, independently of whether the cells were treated with nikkomycin. The authors suggest that the proteins are either integrated into the rigid chitin/chitosan domains or bound to the surface of these lamellar domains (p. 15), but these speculations would be more compelling if the authors could offer some means to distinguish between these options.

It is difficult to determine the exact physical location of these polysaccharide-associated proteins. Our best effort involved providing new data on the water accessibility of the observed protein signals (now provided as **Supplementary Fig. 17b**). The results show that the water-edited intensities (S/S_0) of the protein signals are typically low, ranging from 0.1 to 0.3, which is comparable to those observed for rigid chitin. This indicates that most of the protein signals are likely an integral part of the chitin/chitosan domains rather than residing on the surface, where higher water accessibility and S/S_0 intensities would be expected. We have now explained it in **Lines 443-446**.

Finally, this report includes new observations about constituent-specific hydration based on polarization transfer NMR experiments. Hydration levels were robust for chitosan and poor for both chitin and β -1,3-glucans. The proposed explanations (p. 17) invoked the relative amounts of these constituents, but is it appropriate to consider only the rigid domains? If both rigid and mobile populations must be accounted for, then we need to know their relative proportions to make cogent arguments.

Two changes have been made to address this question. First, we have included a detailed estimate of molecules relative to the full carbohydrate content within the cell wall, considering both rigid and mobile fractions. This change has been described in our response to the first question above (**Lines 326-337**). Second, we have explained that most mobile molecules detected in DP-based experiments are well associated with water, which is essential for maintaining their mobility. In contrast, the rigid molecules detected in CP-based experiments typically originate from those aggregated with fibrillar or partially crystalline domains, such as chitin microfibrils, along with those spatially integrated with them. These molecules have restricted mobility and water accessibility. We have provided an explanation as a new paragraph in **Lines 453-458**.

A related reservation concerns the proposal of “vast highly crystalline domains” described on p. 19 and in Figure 7; do we know what proportion of each constituent is found in experiments that favor the rigid or mobile domain, and that no constituents are double counted in both experiments? Although the authors’ attempt to construct a model for this complex cell-wall system is laudable, the rigor of their reasoning was not always clear to this reader.

We have now provided the estimation of the molecular composition as detailed in our response to previous questions. Additionally, the images presented in **Fig. 8** are hypothetical, mainly for guiding the discussion of the changes induced by nikkomycin, outlined as a new paragraph in **Lines 642-657**.

Reviewer #2

This work uses (primarily) solid-state NMR to study cell walls in five Mucorales species. The paper greatly advances our current knowledge of the cell wall organization both in terms of molecular detail as well as regarding its structural and dynamical properties. Importantly, this information is obtained both before and after treatment with an antifungal inhibitor. However, the authors should clarify several aspects of the research results, their analysis and how they lead towards a unified structural model.

We would like to thank the reviewer for the encouraging comments regarding the usefulness of this study in enhancing our understanding of Mucorales cell walls. We have addressed the questions in a point-by-point manner below to improve clarity regarding both the results and illustrative models.

[1] The paper presents a refined model of the *R. delemar* cell wall organization but does not depict how the cell wall organization is altered after nikkomycin treatment. Since this inhibitor significantly alters the cell wall dimensions and the paper contains highly relevant EM and extensive NMR data of such preparations, such a model should be included in the final figure. For the sake of clarity, it may also be useful to separate results and discussion sections in the text. In addition, it may be useful to comment in a new figure 7 on the location of the different Ch/Cs species that are not found in the current version of Fig. 7.

Thanks. The major change due to the nikkomycin treatment has been better emphasized in the text and is the loss of the β -glucan-chitin-chitosan. This loss induces significant morphogenetic changes and growth inhibition even though this complex is present in a low amount. The role of the β -glucan-chitin in *Rhizopus* is indeed similar to the role displayed by this unique glucan-chitin complex in fungal cell wall in other fungi. We have added a new panel to the model figure (now **Fig. 8b**) to include an illustrative summary of the results observed for nikkomycin-treated cell walls. We have also enhanced **Fig. 8** with additional labels to differentiate between the various molecules, improving clarity. We have significantly expanded and reorganized the Discussion section (now spanning **Lines 552-704**).

The Discussion section is rewritten in a way to ensure that each paragraph has a central focus, together with 20 new references, aiming to improve clarity and better connect with existing literature. The revised Discussion now begins with a paragraph linking to previous knowledge of Zygomycetes cell walls (**Lines 553-562**), followed by two paragraphs connecting NMR-observed chitin/chitosan polymorphism with their biosynthesis (**Lines 564-587**). This is followed by two paragraphs highlighting the novel structural functions of chitosan (**Lines 589-605** and **620-631**) and other polysaccharides (**Lines 633-640**), followed by one paragraph summarizing the changes induced by nikkomycin (**Lines 642-657**), three paragraphs of protein-carbohydrate interactions previously located

in the Results section (**Lines 659-688**), and a concluding paragraph addressing the relevance of these findings to fungal survival in the presence of antifungals (**Lines 690-704**).

We hope these changes can help improve the clarity and make the Discussion section more accessible to our readers.

[2] The data confirm earlier reports that the rigid core of the cell wall mainly comprises chitin and chitosan, with smaller quantities of β -glucans. In addition, the authors speculate about the formation of a β -glucan-chitin/chitosan complex. To the reviewer, the experimental evidence for such an assembly needs further clarifications. Firstly, the NMR data indeed suggest polymorphism with at least 4 different spectroscopic species for chitin and chitosan (a-d). According to lines 261-268, the notion of a β -glucan-chitin/chitosan complex seems to primarily be based on the influence of nikkomycin on chitin/chitosan subspecies 'c' which only contributes a minor part to the entire chitin/chitosan pool (Fig. 3, e). If it is true that nikkomycin treatment almost doubles the cell wall thickness (Fig. 3c), it seems difficult to imagine that the disappearance of such complexes is the only difference in cell wall organization before and after treatment (see comment 1).

We agree with the reviewer that fungal adaptation to stress is complex and involves multiple changes, as observed in several recent studies. Following the advice, we have now included a new **Fig. 8b** to summarize the changes in cell wall structure and molecular distribution after nikkomycin exposure. Although not to scale, this figure serves as a guide to explain the key changes which have been defined only with these biophysical methodologies. A new paragraph in **Lines 642-657** highlights the molecular-level changes, while the alterations in cell wall thickness, water retention, and rigidity are reiterated in **Lines 690-696** and linked to earlier observations of similar survival strategies in fungi.

Along similar lines, Page 7/ Figure 7 suggest that a significant portion of chitin and chitosan exist as domains within copolymers. Notably, the NHN experiment (fig. 1e) contains only one rather weak cross peak and uses long mixing times that may compromise a detailed structural analysis. In addition, Lysine (observed in the amino-acid analysis) side-chain signals are found at similar ^{15}N chemical-shift values. The claim (lines 186-188) that “The observed sub-nanometer correlations between amide and amine groups ... provide additional evidence supporting the idea that a significant portion of chitin and chitosan exist as domains within copolymers.” needs further clarification.

We concur with the reviewer that, based on the hNH spectra alone, it is challenging to definitively confirm whether the long-range cross peak is solely from chitin-chitosan interactions or if it also includes contributions from lysine. To address this, we have provided further explanations and additional experimental data in the new **Fig. 2c**. First, we have combined the NC and NH HETCOR spectra in **Fig. 2a** to demonstrate the alignment of chitin/chitosan signals with amide/amine signals for confirmation purposes. Second, we conducted a 2D hChH (RFDR) experiment to show the correlations between chitin/chitosan carbons and the chitin amide H^{N} . Given that proteins are relatively low in content in these cell-wall-rich samples, the expected protein- H^{N} cross peaks are not observed for ^{13}C chemical shifts of 10-50 ppm, except for the strong cross peaks between chitin methyl carbon (ChMe) with carbohydrate protons. These new data confirm the presence of sub-nanometer chitin-chitosan

domains in the *R. delemar* cell wall. We have now clarified these points as a separate paragraph in **Lines 199-211**.

[3] To probe the supramolecular assembly, the authors mainly use long NMR mixing times. The spectrum (Figure 4a) contains mostly broad correlations that are often difficult to attribute to individual peaks seen in the “short-range” experiment. Even in spectral regions that only contain one or a limited set of “shot-range” correlations (e.g. around 78 ppm, 100 ppm, 25 ppm) broad, featureless peaks are found. It would help if the authors (possibly in the SI) exemplify some of the additional correlations using vertical/horizontal dashed lines in the spectrum. In addition, they report in the SI (SI Figure 10 & caption) that a short (15 ms) PAR spectrum only retained chitin signal, “revealing the disordered nature of b-glucan and chitosan signals...” If this is the case, would the latter observations not speak against the co-existence of chitin-chitosan fibers as suggested in the model of Figure 7?

Thanks for the helpful advice. We have now added 1D cross sections of long-range correlation spectra as new **Supplementary Figures 11 and 13c** to exemplify these long-range cross peaks. The broad, featureless peaks are mainly due to the significant structural heterogeneity of these macromolecules that we need to handle in their native cellular environment, where we have to stare at the fine features of the signals instead of having purely resolved cross peaks like observed in protein NMR. Chitin is the most crystalline component in the cell wall, surviving through the ^1H and ^{13}C relaxations during the 15 ms PAR duration. Chitosan and β -glucans, although associated with chitin, accommodate local structural disorders and could not survive through PAR. We now provided a clarification in **Lines 369-373** and the captions of **Supplementary Fig. 12**.

[4] The authors manage to identify a number of amino-acid signals and report chemical-shift assignments (table S5). In addition, they should include an analysis of these assignments in terms of protein secondary structure for both (rigid & mobile) protein species. Also, further information about the relative abundance of the different amino acids would be helpful. Can such information be correlated to polypeptides/proteins known to be associated with Mucorales cell walls? The abstract also claims that “some proteins are entrapped within this semi-crystalline chitin/chitosan layer, stabilized by the sidechains of hydrophobic amino acid residues, and situated distantly from β -glucans”. This claim seems to be mainly based on Fig. 4d/e which reveals weak correlations between the amino-acid methyl carbons and Cs/Ch NMR signals. Please indicate this spectral region in Figure 4a and comment whether also Ca/Cb correlations to Ch/Cs occur. Did the authors observe similar correlations for other “rigid” amino acids? If not, please explain.

Thanks. We have now included a new **Supplementary Fig. 16** to show the secondary structure of the proteins in *R. delemar* cell walls. Additionally, the relative abundance of major amino acid types observed in the rigid and mobile fractions is presented in a new **Supplementary Fig. 17a**. We have now mentioned it in **Lines 412-415 and 805-806**.

Proteins in the Mucorales mycelial cell walls are highly varied, with no dominant protein structure identified so far. This contrasts with conidia rodlets (Zygomycete conidia also have rodlets which are not present in the mycelia studied here), where proteins often exhibit repeated domains/structures and

can form distinct layers. We consulted several mycologists regarding this possibility, but it turned out to be not feasible to specifically correlate the NMR spectra with protein structures. However, possible chitin/chitosan-binding proteins in fungi and other organisms were mentioned in the Discussion section (**Lines 670-678**).

A dashed line box was added to (now) **Figure 5a** to show the spectral region for the zoomed-in view of panel b. We did not observe clear cross peaks between the C α /C β sites and chitin/chitosan signals, likely due to these sites being embedded within the protein structure. Additionally, the weak C α signals are overlapped by the dominant C2 signals from chitin/chitosan. No similar correlations were observed for other rigid amino acids, suggesting the prevalent involvement of isoleucine in the protein-carbohydrate interface specifically in *Rhizopus* species. We have now explained these aspects in **Lines 421-425**.

[5] Minor points:

- Lines 131-133: It is not clear what the authors mean. ¹H detection on fungal cell walls has already been demonstrated before. Please clarify or remove. Also, they write “For example, the ¹H chemical shift of chitin H1 (Ch1) spanned the range of 3.2-5.2 ppm encompassing at least 8 distinguishable features..” Please indicate those in the figure. To the reviewer, the corresponding cross peak region is rather featureless which may indeed reflect chitin chain heterogeneity as regularly seen in MAS NMR of biopolymers. The same applies to the NC chitin/chitosan correlations.

Thanks. These were short introductory sentences for our readers without NMR background. We have revised the introductory sentences to be more concise and have rearranged their placement in the paragraph to enhance coherence (**Lines 129-133**). We also included more recent references (e.g., Vallet et al., JMR 2024; Duan and Hong, JPCB 2024; Bahri et al., JBNMR 2023). The eight resolvable features of chitin signals are now highlighted with arrows in Fig. 1c, and this has been noted in the captions. We also appreciate the comment regarding structural heterogeneity. We now mention it as structural heterogeneity or complexity in **Line 126, 133 and 141**.

- NMR Chemical shift signatures of alpha- and beta-chitin structure have been published and are cited in this work. To the reviewer, it would be useful to compare the current assignments to these literature values in more detail. The chemical shift variation reported for the different Ch,Cs species (table S3) is often rather small (0.1-0.3 ppm) which probably is smaller than the actual ¹³C line width. Please clarify which of the variations are actually spectroscopically significant and whether those cannot be attributed to local changes such as ¹H-bonding or molecular packing.

Two changes were made to better present the chemical shift analysis. First, heatmap analysis comparing chitin and chitosan chemical shifts with model structures and between different *Rhizopus* and *Mucor* species are now included as new **Fig. 7d** and **Supplementary Fig. 24**, and the paragraph in **Lines 540-550**. Second, we provide more detailed information on the chemical shift assignment by expanding the paragraph in **Lines 154-169** and better connecting it with Supplementary Fig. 2, which shows the resolved carbon connectivity for each chitin and chitosan form. We also emphasize that we mainly relied on the 2D cross peaks for resolving these forms instead of comparing chemical shifts for each carbon site.

Typographical error: Line 276: constitute

Corrected.

Edits in Response to Editorial Requests

* Please replace your bar graphs with plots that feature information about the distribution of the underlying data. All data points should be shown for plots with a sample size less than 10. For larger sample sizes, please consider box-and-whisker or violin plots as alternatives. Measures of centrality, dispersion and/or error bars should be plotted and described in the figure legend.

We have replaced the bar graph in (now) **Fig. 6a** to box and whisker plot. We also removed the bar graphs in **Fig. 6b, c** and showed all the data points since the sample size is less than 10 for these two plots. We also added clarification of the plots as well as the sample size (n) in the figure captions.

REVIEWERS' COMMENTS

Reviewer #1 (Remarks to the Author):

The authors have responded to the reviewer comments constructively and thoroughly, adding clarifications as well as new data and analysis. They have also acknowledged several challenges encountered in this study, which speaks to their honesty and can be helpful to their audience. In my view, this excellent manuscript is now ready for publication.

Reviewer #2 (Remarks to the Author):

The authors have done an excellent job in addressing the issues raised earlier. The modifications and additions to the figures and text further strengthen the scientific conclusions and relevance of this work. I recommend publication of this paper as is.

Responses to Reviewers' Comments

Response: We appreciate the positive comment from both reviewers.